# Coupling adaptive molecular evolution to phylodynamics using fitness-dependent birth-death models

David A Rasmussen[1,2]*, Tanja Stadler[3,4]

[1]Department of Entomology and Plant Pathology, North Carolina State University, Raleigh, United States; [2]Bioinformatics Research Center, North Carolina State University, Raleigh, United States; [3]Department of Biosystems Science and Engineering, ETH Zürich, Basel, Switzerland; [4]Swiss Institute of Bioinformatics, Lausanne, Switzerland

**Abstract** Beneficial and deleterious mutations cause the fitness of lineages to vary across a phylogeny and thereby shape its branching structure. While standard phylogenetic models do not allow mutations to feedback and shape trees, birth-death models can account for this feedback by letting the fitness of lineages depend on their type. To date, these multi-type birth-death models have only been applied to cases where a lineage's fitness is determined by a single character state. We extend these models to track sequence evolution at multiple sites. This approach remains computationally tractable by tracking the genotype and fitness of lineages probabilistically in an approximate manner. Although approximate, we show that we can accurately estimate the fitness of lineages and site-specific mutational fitness effects from phylogenies. We apply this approach to estimate the population-level fitness effects of mutations in Ebola and influenza virus, and compare our estimates with in vitro fitness measurements for these mutations.
DOI: https://doi.org/10.7554/eLife.45562.001

*For correspondence:
drasmus@ncsu.edu

Competing interests: The authors declare that no competing interests exist.

## Introduction

The fitness effects of new mutations is a key determinant of a population's evolutionary potential to adapt over time. Studies exploring the distribution of fitness effects (DFE) in a wide range of organisms have revealed that, while many mutations are neutral, a smaller but significant fraction have substantial effects on fitness (*Sanjuán et al., 2004*; *Eyre-Walker and Keightley, 2007*; *Visher et al., 2016*). These findings have spurred interest in molecular evolutionary models that consider how non-neutral mutations shape sequence evolution and patterns of genetic diversity. Such models range in complexity from simple models assuming that selection operates uniformly across all sites (*Muse and Gaut, 1994*; *Goldman and Yang, 1994*; *Yang and Nielsen, 2008*) to parameter rich models with site-specific fitness effects (*Halpern and Bruno, 1998*; *Lartillot and Philippe, 2004*; *Rodrigue et al., 2010*; *Hilton and Bloom, 2018*). While all of these models assume sequences evolve along an underlying phylogenetic tree representing their shared common ancestry, all also assume that the mutation process driving sequence evolution is independent of the other evolutionary processes giving rise to the tree. This independence assumption implies that mutations do not feedback and affect the fitness of lineages in the tree, such that lineages carrying highly beneficial mutations are just as likely to survive and produce sampled descendants as lineages riddled with deleterious mutations (*Figure 1A*).

While questionable in terms of biological realism, independence between the tree generating process and the mutation process allows for tractable statistical models. Assuming independence,

**Figure 1.** Schematic overview of birth-death models. (**A**) Standard phylogenetic models assume that there is an underlying process by which individuals replicate and give rise to a phylogeny. Mutations occur along the lineages of the tree, generating the sequence data observed at the tips. The mutation process is assumed to be independent of tree generating process, such that mutations do not impact the branching structure of the tree. (**B**) The MFBD allows us to relax this assumption, such that mutations at multiple sites feedback and shape both the tree and sequence data. (**C**) Under the original multi-type birth-death model we track $D_{n,i}(t)$, the probability density that a lineage $n$ at time $t$ in state $i$ produces the subtree descending from $n$ and the observed tip states. We also track $E_i$, the probability that a lineage produces no sampled descendants and is therefore unobserved. (**D**) In the MFBD model we instead track $D_{n,k,i}(t)$, the probability that a lineage $n$ in state $i$ at site $k$ produces the subtree and the observed tip states at site $k$. Because the fitness of a lineage $f_n$ will depend on its genotype at all sites, we use the marginal site probabilities $\omega$ to compute the probability that a lineage has a certain genotype, such as ACT (Approximation 1). We can then marginalize over the fitness of each genotype weighted by its approximate genotype probability to compute the fitness $f_n$ of a lineage (Approximation 2). Finally, we need to know the probability $E_n$ that a lineage left no other sampled descendants, which we approximate using the probability $E_u$ that a lineage with same expected fitness $u$ leaves no sampled descendants (Approximation 3). The schematic in A was reproduced from the original figure by Louis du Plessis (https://github.com/Taming-the-BEAST/TechnicalLectureSources/tree/master/BeastIntro2018) with permission under a Creative Commons license.

DOI: https://doi.org/10.7554/eLife.45562.002

the joint likelihood of a phylogenetic tree $\mathcal{T}$ and the sequence data $\mathcal{S}$ at the tips of the tree having evolved as observed can be factored into two distinct components:

$$L(\mathcal{S}, \mathcal{T}|\mu, \theta) = L(\mathcal{S}|\mathcal{T}, \mu)p(\mathcal{T}|\theta). \tag{1}$$

The likelihood of the sequence data $L(\mathcal{S}|\mathcal{T}, \mu)$ conditional on the tree and the mutational parameters $\mu$ can be computed efficiently for most continuous-time Markov models of sequence evolution (**Felsenstein, 1981**). The probability density $p(\mathcal{T}|\theta)$ of the tree $\mathcal{T}$ given the parameters generating the tree $\theta$ can likewise be computed under widely used coalescent (**Griffiths and Tavaré, 1994**; **Pybus et al., 2000**) or birth-death models (**Rannala and Yang, 1996**; **Stadler, 2009**). In Bayesian phylogenetics, $p(\mathcal{T}|\theta)$ is normally thought of as the prior distribution over trees rather than a likelihood, because the tree itself is inferred from the sequence data.

The assumption of independence between the mutation and tree generating processes may be unproblematic in certain scenarios, such as if mutations are truly neutral or do not contribute to substantial fitness differences among lineages. A common argument invoked in defense of ignoring non-neutral mutations is that macroevolutionary tree generating processes like speciation and extinction play out on longer timescales than the substitution process fixing or removing mutations within a population (*Bustamante, 2005*). In this case, fitness variation drives the substitution process within a population but does not ultimately drive the formation of a phylogeny at the species level. But such separation-of-timescales arguments do not hold when segregating mutations contribute to substantial fitness variation between lineages in a phylogeny, such as for rapidly evolving microbes where several different mutant strains can co-circulate. In these cases, the tree generating and mutation processes occur on the same timescale, and the fitness effects of mutations can feedback and shape the branching structure of a phylogeny (*Kaplan et al., 1988*; *Nicolaisen and Desai, 2012*; *Neher and Hallatschek, 2013*). Ignoring non-neutral evolution in this case may introduce biases into phylogenetic inference. But perhaps more importantly, fitness differences among lineages can be correlated with ancestral genotypes, providing information about the molecular basis of adaptive evolution we would otherwise ignore.

We therefore explore an approach that couples molecular sequence evolution to the tree-generating process using multi-type birth-death (MTBD) models. Under this approach, mutations can directly impact the fitness of a lineage in the phylogeny by altering its birth or death rate (*Figure 1B*). For a single evolving site or other character state, the joint likelihood of the phylogeny together with the observed tip states can be computed exactly under the MTBD model (*Maddison et al., 2007*; *Stadler and Bonhoeffer, 2013*; *Kühnert et al., 2016*). However, this approach is impractical for more than a few non-neutrally evolving sites due to the need to track all possible genotypes as separate types in the state space of the model. We therefore explore an approximate birth-death model that considers how mutations at multiple sites contribute to a lineage's overall fitness, without the need to track all possible genotypes in sequence space. This approach allows us to infer the fitness effects of individual mutations and the fitness of any particular lineage at any time (based on its inferred ancestral genotype) from the branching structure of a phylogeny. Because our approach is particularly relevant to rapidly adapting microbial pathogens, we apply it to Ebola and influenza virus sequence data in order to quantify the fitness effects of naturally occurring amino acid substitutions.

## Materials and methods

### The MTBD at a single evolving site

At a single evolving site, the multi-type birth-death (MTBD) model of *Stadler and Bonhoeffer (2013)* can be used to compute the joint likelihood $L(S, \mathcal{T}|\mu, \theta)$ of the sequence or character state data $S$ and phylogenetic tree $\mathcal{T}$ in a way that couples the mutation process with changes in fitness along a lineage. Let $D_n(t)$ represent the probability density (i.e. the likelihood) that the subtree descending from lineage $n$ evolved between time $t$ and the present exactly as observed (*Figure 1C*). Further, let $D_{n,i}(t)$ represent this probability density conditional on lineage $n$ being in state $i$ out of $M$ possible states at time $t$. Here the state of a lineage refers to a particular allele or character state (e.g. nucleotide or amino acid) at a single site. We reserve the term genotype to refer to a particular configuration of states across multiple sites in a sequence.

The density $D_{n,i}(t)$ can be computed going backwards in time from the present ($t = 0$) to time $t$ along a lineage by numerically solving a system of ordinary differential equations:

$$
\begin{aligned}
\frac{d}{dt}D_{n,i}(t) \; = \; & -(\lambda_i + \sum_{j=1}^{M}\gamma_{i,j} + d_i)D_{n,i}(t) \quad \text{(a) no event} \\
& + 2\lambda_i E_i(t)D_{n,i}(t) \qquad \text{(b) birth of lineage with no sample descendants} \\
& + \sum_{j=1}^{M}\gamma_{i,j}D_{n,j}(t) \qquad \textit{(c) mutation from i to j}
\end{aligned}
\tag{2}
$$

Here, $\lambda_i$ is the birth rate and $d_i$ is the death rate of lineages in state $i$, and thus reflect a lineage's fitness. Mutations between states $i$ and $j$ occur at a rate $\gamma_{i,j}$, independently of birth events. Each

term in (*Equation 2*) describes how $D_{n,i}$ changes through time by accounting for all of the different events that could have occurred along the lineage. The first term (a) considers the change in probability density given that no birth, death or mutation event occurred. The second term (b) considers the probability of a birth event that went unobserved because one of the child lineages produced no sampled descendants (this event has probability $E_i(t)$, see below). The third term (c) reflects the probability that the lineage mutated from state $i$ to $j$.

$E_i(t)$ represents the probability that a lineage in state $i$ is not sampled and has no sampled descendants. This probability can be computed at any time $t$ by solving a second set of ODEs:

$$
\begin{aligned}
\frac{d}{dt}E_i(t) = {} & (1-s_i)d_i && \text{(a) death without sampling} \\
& -(\lambda_i + \sum_{j=1}^{M}\gamma_{i,j} + d_i)E_i(t) && \text{(b) no event} \\
& +\lambda_i E_i(t)^2 && \text{(c) birth, neither child has sampleled descendants} \\
& +\sum_{j=1}^{M}\gamma_{i,j}E_j(t). && \text{(d) mutation from i to j}
\end{aligned} \tag{3}
$$

The first term (a) reflects the probability that a lineage dies and is not sampled, where $s_i$ is the probability that a lineage in state $i$ is sampled upon dying. Terms b-d have similar interpretations as in (*Equation 2*).

At a tip lineage $n$, we initialize $D_{n,i}(t) = d_i s_i$ if the lineage was sampled upon death at time $t$. Alternatively, if $n$ was sampled at the present time $t = 0$ before dying, then $D_{n,i}(t) = \rho_i$, where $\rho_i$ is the probability that an individual in state $i$ was sampled at present. At a branching event, the probability density $D_{a,i}$ of the parent lineage $a$ in state $i$ giving rise to two descendent lineages $n$ and $m$ is updated as:

$$
D_{a,i} = 2\lambda_i D_{m,i}(t)D_{n,i}(t). \tag{4}
$$

The factor of two enters because either lineage $m$ or $n$ could have given birth and we must consider both possible events.

At the root, we can compute the probability density of the entire tree by summing over all possible root states:

$$
D_n = \sum_{i=1}^{M} q_i \frac{D_{n,i}(t_{root})}{1 - E_i(t_{root})}, \tag{5}
$$

where $q_i$ is the prior probability that the root is in state $i$ at time $t_{root}$. Including the term $1 - E_i(t_{root})$ in the denominator conditions the birth-death process on giving rise to at least one sampled individual. $D_n$ represents the probability that the entire tree and the tip states $\mathcal{S}$ evolved as exactly as observed. It is therefore equivalent to the joint likelihood $L(\mathcal{S},\mathcal{T}|\mu,\theta)$ we seek where $\mu = \{\gamma\}$ and $\theta = \{\lambda,d,s\}$.

In theory, this approach could be extended to evolution at any number of sites as long as we track $D_{n,i}(t)$ for all possible genotypes $i$. Unfortunately, this approach has limited utility because the number of possible genotypes in sequence space scales exponentially with the number of sites $L$ (i.e. $4^L$ possible genotypes for nucleotide sequences), making the MTBD model impractical for modeling evolution at more than a few sites.

## The marginal fitness birth-death model

While the fitness of a lineage will generally depend on its genotype across multiple sites, tracking evolution in the space of all possible genotypes is, as just discussed, computationally infeasible. We therefore seek an approach that considers how mutations at multiple sites determine the fitness of a lineage without the need to track $D_{n,i}$ for all possible genotypes. In the approach described below and outlined in *Figure 1D*, we therefore track molecular evolution at each site, computing the probability that each site occupies each state, and then approximate the probability of a lineage being in any particular genotype based on these site probabilities. To compute the expected fitness of a lineage, we can then sum, or marginalize, over the fitness of each genotype weighted by its

approximate probability. We therefore refer to this approach as the marginal fitness birth-death (MFBD) model.

First, in order to couple a lineage's fitness with the birth-death process, we will assume that the birth rate $\lambda_n$ of any lineage $n$ scales according to the fitness $f_g$ of its genotype:

$$\lambda_n = f_g \lambda_0, \tag{6}$$

where $\lambda_0$ is the base birth rate assigned to a particular reference genotype (e.g. the wildtype). A lineage's death rate can also be coupled to its fitness, but for simplicity we will assume a lineage's fitness is reflected only in its birth rate $\lambda_n$.

Let $\mathcal{G}$ be the set of all possible genotypes in sequence space and $g_k$ be the state of genotype $g$ at site $k$. To make it clear when we are considering evolution in genotype space rather than at a particular site, we will write the probability density $D_{n,i}$ as $D_{n,g}$ when $i$ refers to a particular genotype. Furthermore, let $D_{n,k,i}$ be the probability density of the subtree descending from lineage $n$ given that site $k$ is in state $i$. By definition,

$$D_{n,k,i} = \sum_{\{g \in \mathcal{G}: g_k = i\}} D_{n,g}, \tag{7}$$

where the sum is over all genotypes in $\mathcal{G}$ with site $k$ in state $i$.

We can derive a difference equation for $D_{n,k,i}$ from $D_{n,g}$ in a straightforward manner:

$$
\begin{aligned}
D_{n,k,i}(t + \Delta t) &= \sum_{\{g \in \mathcal{G}: g_k = i\}} D_{n,g}(t + \Delta t) \\
&= \sum_{\{g \in \mathcal{G}: g_k = i\}} \Big[ \big(1 - (f_g \lambda_0 + \sum_{j=1}^{M} \sum_{\{g' \in \mathcal{G}: g'_k = j\}} \gamma_{g,g'} + d)\big) D_{n,g}(t) \Delta t \\
&= + 2 f_g \lambda_0 E_{n,g}(t) D_{n,g}(t) \Delta t \\
&= + \sum_{j=1}^{M} \sum_{\{g' \in \mathcal{G}: g'_k = j\}} \gamma_{g,g'} D_{n,g'}(t) \Delta t \Big].
\end{aligned}
\tag{8}
$$

Taking the limit as $\Delta t \to 0$, we get a new system of differential equations for $D_{n,k,i}(t)$:

$$
\begin{aligned}
\frac{d}{dt} D_{n,k,i}(t) &= \sum_{\{g \in \mathcal{G}: g_k = i\}} \Big[ -(f_g \lambda_0 + \sum_{j=1}^{M} \sum_{\{g' \in \mathcal{G}: g'_k = j\}} \gamma_{g,g'} + d) D_{n,g}(t) \\
&\quad + 2 f_g \lambda_0 E_{n,g}(t) D_{n,g}(t) \\
&\quad + \sum_{j=1}^{M} \sum_{\{g' \in \mathcal{G}: g'_k = j\}} \gamma_{g,g'} D_{n,g'}(t) \Big]
\end{aligned}
$$

Unfortunately, (*Equation 9*) would still require us to track $D_{n,g}(t)$ for all possible genotypes, precisely what we wish not to do. We show below that, if we can approximate $f_g$ and $E_{n,g}$ for any given lineage, we can write (*Equation 9*) in terms of only $D_{n,k,i}$ (see (*Equation 19*)) and therefore do not need to track each genotype.

## Approximating the fitness of a lineage

We begin by approximating the fitness $f_n$ of a lineage $n$. Even if we do not know the exact genotype of a lineage at a particular time, we can compute the lineage's expected fitness by summing over the fitness of each genotype $f_g$ weighted by the probability $\omega_{n,g}$ that lineage $n$ is in genotype $g$:

$$\mathbb{E}(f_n) = \sum_{g \in \mathcal{G}} f_g \omega_{n,g}. \tag{10}$$

The same logic can be extended to compute the expected marginal fitness $\mathbb{E}(f_{n,k,i})$ of a lineage $n$ that at site $k$ is in state $i$:

$$\mathbb{E}(f_{n,k,i}) = \sum_{\{g \in \mathcal{G}: g_k = i\}} f_g \omega_{n,g}. \tag{11}$$

Computing $\mathbb{E}(f_{n,k,i})$ using (*Equation 11*) requires knowledge of the genotype probabilities $\omega_{n,g}$, which would again require us to track evolution in genotype space. We therefore introduce our major assumption: that we can approximate genotype probabilities using only the marginal site probabilities $\omega_{n,k,i}$ that site $k$ is in state $i$. We describe how we compute $\omega_{n,k,i}$ below. For now, we make the approximation that

$$\hat{\omega}_{n,g} = \frac{\prod_{k=1}^{L} \omega_{n,k,g_k}}{\sum_{g \in \mathcal{G}} \prod_{k=1}^{L} \omega_{n,k,g_k}}. \tag{12}$$

This approximation assumes that all sites evolve independently of one another, which is not generally true because mutations at different sites are linked together in genotypes with shared ancestral histories, creating correlations among sites that we ignore.

Using the approximate genotype probabilities $\hat{\omega}_{n,g}$, we can in turn approximate the expected marginal fitness of a lineage:

$$\hat{f}_{n,k,i} = \sum_{\{g \in \mathcal{G}: g_k = i\}} f_g \hat{\omega}_{n,g}. \tag{13}$$

If the fitness effects of each site act multiplicatively to determine the overall fitness of a lineage, we can compute $\hat{f}_{n,k,i}$ as:

$$\hat{f}_{n,k,i} = \sigma_{ki} \prod_{l=1, l \neq k}^{L} \sum_{j=1}^{M} \sigma_{lj} \omega_{n,l,j}, \tag{14}$$

where $\sigma_{ki}$ is the fitness effect of site $k$ being in state $i$. This formulation of $\hat{f}_{n,k,i}$ is useful if the number of sites $L$ is large and the number of genotypes we need to sum over in (*Equation 13*) is therefore also extremely large.

## Approximating the probability of no sampled descendants

The $E_{n,g}(t)$ term in (*Equation 9*) represents the probability that a lineage $n$ alive at time $t$ in the past is not sampled and leaves behind no sampled descendants. $E_{n,g}(t)$ therefore necessarily depends on the fitness of unobserved lineages descending from $n$ and how fitness along these lineages evolves through changes in their genotype. Because it is often easier track evolution in one dimensional fitness space rather than high-dimensional sequence space (*Kepler and Perelson, 1993*; *Tsimring et al., 1996*), we simplify this problem by tracking a proxy for $E_{n,g}(t)$ though fitness space.

Let $E_u$ be the probability that a lineage with expected fitness $u$ leaves no sampled descendants. While fitness can take on a continuous range of values, we track these probabilities only for a discrete set of points $\mathcal{V}$ in fitness space. We can track $E_u$ for $u \in \mathcal{V}$ by modifying (*Equation 3*) to obtain:

$$\frac{d}{dt} E_u(t) = (1 - s_u) d_u - (\lambda_u + \sum_{v \in \mathcal{V}} \gamma_{u,v} + d_u) E_u(t) + \lambda_u E_u(t)^2 + \sum_{v \in \mathcal{V}} \gamma_{u,v} E_v(t). \tag{15}$$

We can then substitute $E_{n,g}(t)$ in (*Equation 9*) with $E_u$ for the fitness value $u$ closest to $f_g$ or $\hat{f}_{n,k,i}$ in fitness space.

Tracking evolution in fitness space requires us to specify rates $\gamma_{u,v}$ for how lineages transition between fitness classes $u$ and $v$. Let $\mathcal{G}_u$ be the set of genotypes with expected fitness closest to $u$ out of all fitness values in $\mathcal{V}$. We approximate $\gamma_{u,v}$ as:

$$\gamma_{u,v} = \frac{1}{|\mathcal{G}_u|} \sum_{i \in \mathcal{G}_u} \sum_{j \in \mathcal{G}_v} \mu_{ij}, \tag{16}$$

where $\mu_{ij}$ is the mutation rate between genotypes $i$ and $j$. In other words, we compute the average rate of transitions out of fitness class $u$ into $v$ by summing over all possible transitions between genotypes contained within each fitness class. Note that if each genotype falls in a unique fitness class such that $|\mathcal{G}_u| = 1$ for all $u \in \mathcal{V}$, then $E_u$ is computed exactly. In the Results, we compare using the approximate transition rates above to compute $E_u$ versus an even simpler approximation where we assume no transitions between fitness classes along unobserved lineages, which has been

assumed in earlier multi-type birth-death models (*Rabosky et al., 2014*; *Barido-Sottani et al., 2018*).

### Computing the marginal site densities $D_{n,k,i}$

Recall that (*Equation 9*) provided an exact way to track the marginal site densities $D_{n,k,i}$ based on the genotype densities $D_{n,g}$. To efficiently evaluate $D_{n,k,i}$ without the need to track $D_{n,g}$ for all genotypes, we apply the three approximations made above. First, we approximate the genotype probabilities $\hat{\omega}_{n,g}$ based on the marginal site probabilities. Second, we marginalize over the fitness of each genotype (weighted by its genotype probability) to compute $\hat{f}_{n,k,i}$ and then substitute $\hat{f}_{n,k,i}$ for $f_g$ for all genotypes where $g_k = i$ below. Third, we approximate $E_{n,g}$ by $E_u$ for a single fitness value $u$ closest to $\hat{f}_{n,k,i}$. Making these approximations in (*Equation 9*) leads to:

$$
\begin{aligned}
\frac{d}{dt}D_{n,k,i}(t) = &\sum_{\{g\in\mathcal{G}:g_k=i\}}[-(\hat{f}_{n,k,i}\lambda_0+\sum_{j=1}^{M}\sum_{\{g'\in\mathcal{G}:g'_k=j\}}\gamma_{g,g'}+d)D_{n,g}(t)\\
&+2\hat{f}_{n,k,i}\lambda_0 E_u(t)D_{n,g}(t)\\
&+\sum_{j=1}^{M}\sum_{\{g'\in\mathcal{G}:g'_k=j\}}\gamma_{g,g'}D_{n,g'}(t)]
\end{aligned}
\tag{17}
$$

Assuming that the mutation rate from $i$ to $j$ at site $k$ does not depend on the genetic background, we can substitute $\sum_{j=1}^{M}\sum_{\{g'\in\mathcal{G}:g'_k=j\}}\gamma_{g,g'}$ with $\sum_{j=1}^{M}\gamma_{i,j}$, where $\gamma_{i,j}$ is the per site mutation rate. We can likewise substitute $\sum_{j=1}^{M}\sum_{\{g'\in\mathcal{G}:g'_k=j\}}\gamma_{g,g'}D_{n,g'}(t)$ with $\sum_{j=1}^{M}\gamma_{i,j}\sum_{\{g'\in\mathcal{G}:g'_k=j\}}D_{n,g'}(t)$. Making these substitutions and rearranging the sums in (*Equation 17*), we have:

$$
\begin{aligned}
\frac{d}{dt}D_{n,k,i}(t) = &-(\hat{f}_{n,k,i}\lambda_0+\sum_{j=1}^{M}\gamma_{i,j}+d)\sum_{\{g\in\mathcal{G}:g_k=i\}}D_{n,g}(t)\\
&+2\hat{f}_{n,k,i}\lambda_0 E_u(t)\sum_{\{g\in\mathcal{G}:g_k=i\}}D_{n,g}(t)\\
&+\sum_{j=1}^{M}\gamma_{i,j}\sum_{\{g'\in\mathcal{G}:g'_k=j\}}D_{n,g'}
\end{aligned}
\tag{18}
$$

Recalling that $D_{n,k,i}=\sum_{\{g\in\mathcal{G}:g_k=i\}}D_{n,g}$ (and by extension $D_{n,k,j}=\sum_{\{g\in\mathcal{G}:g_k=j\}}D_{n,g}$), we have:

$$
\begin{aligned}
\frac{d}{dt}D_{n,k,i}(t) = &-\left(\hat{f}_{n,k,i}\lambda_0+\sum_{j=1}^{M}\gamma_{i,j}+d\right)D_{n,k,i}(t)\\
&+2\hat{f}_{n,k,i}\lambda_0 E_u(t)D_{n,k,i}(t)\\
&+\sum_{j=1}^{M}\gamma_{i,j}D_{n,k,j}(t).
\end{aligned}
\tag{19}
$$

The significance of (*Equation 19*) is twofold. First, we can track sequence evolution at each site individually without tracking all genotypes. Second, given $\hat{f}_{n,k,i}$, we can track the overall fitness of a lineage by marginalizing over the fitness effects of all possible mutations at other sites. We can therefore track sequence evolution at each site while simultaneously taking into account the coupled fitness effects of mutations at all other sites on a lineage's fitness.

Computing $\hat{f}_{n,k,i}$ still requires us to approximate the genotype probabilities using (*Equation 12*), which in turn requires the marginal site probabilities $\omega_{n,k,i}$. In our notation, $\omega_{n,k,i}$ represents the conditional probability $p(i|\mathcal{T}_n,\mathcal{S}_n)$ that lineage $n$ is in particular state $i$, where $\mathcal{T}_n$ represents the subtree descending from $n$ with tip sequences $\mathcal{S}_n$ represents the inverse conditional probability density $p(\mathcal{T}_n,\mathcal{S}_n|i)$. We can therefore apply Bayes theorem to compute $\omega_{n,k,i}$ given $D_{n,k,i}$:

$$
\omega_{n,k,i}=p(i|\mathcal{T}_n,\mathcal{S}_n)=\frac{p(\mathcal{T}_n,\mathcal{S}_n|i)q(i)}{\sum_i^M p(\mathcal{T}_n,\mathcal{S}_n|i)q(i)}=\frac{D_{n,k,i}q(i)}{\sum_i^M D_{n,k,i}q(i)}.
\tag{20}
$$

The $q(i)$ terms represent the prior probability that the lineage is in state $i$. Here we make a simplification in assuming that the tree ancestral and sister to lineage $n$ has no information regarding $\omega_{n,k,i}$, and thus assume a uniform prior on $q(i) = 1/M$. The $q(i)$ terms therefore cancel above.

Because the fitness of a lineage depends on the state of all sites, we must solve (*Equation 19*) for all sites simultaneously as one coupled system of differential equations. This requires updating $D_{n,k,i}$ at each time step, which suggests the following iterative procedure.

At a tip $n$ observed to be in genotype $g$, we initialize $\hat{f}_{n,k,i}$ as $f_g$ if $g_k = i$ or else $\hat{f}_{n,k,i} = 0$, $\hat{D}_{n,k,i} = ds$ or $\rho$, and $\omega_{n,k,i} = 1$ if $g_k = i$, else $\omega_{n,k,i} = 0$. Then at each time step backwards through time from time $t$ to time $t + \Delta t$, for each site and state we:

1. Update $D_{n,k,i}$ by numerically integrating (*Equation 19*) over time step $\Delta t$.
2. Update the marginal site probabilities $\omega_{n,k,i}$ using (*Equation 20*)
3. Update the expected marginal fitness values $\hat{f}_{n,k,i}$ using (*Equation 13*) or (*Equation 14*).

## Computing the full joint likelihood

We can now compute the joint likelihood of the tree and sequence data if we track $D_{n,k,i}$ at each site back to the root. At the root, $D_{n,k,i}(t_{root})$ represents $p(\mathcal{T}, \mathcal{S}_k | \mu, \theta, i)$, the probability density of the entire tree $\mathcal{T}$ and the observed sequence data $\mathcal{S}_k$ as site $k$, conditional on site $k$ being in state $i$ at the root. To be precise, $D_{n,k,i}$ only approximates $p(\mathcal{T}, \mathcal{S}_k | \mu, \theta, i)$ because we computed $D_{n,k,i}$ using the expected marginal fitness of a lineage $\hat{f}_{n,k,i}$ based on approximate genotype probabilities. We therefore introduce an additional auxiliary variable $\mathcal{F}$ representing the entire set of expected fitness values $\hat{f}_{n,k,i}$ computed over all lineages, sites and states. Using this notation, $D_{n,k,i}(t_{root}) = p(\mathcal{T}, \mathcal{S}_k | \mu, \theta, \mathcal{F}, i)$. By summing over all possible root states at site $k$ (and conditioning on survival), we can then compute:

$$p(\mathcal{T}, \mathcal{S}_k | \mu, \theta, \mathcal{F}) = \sum_{i=1}^{M} q(i) \frac{p(\mathcal{T}, \mathcal{S}_k | \mu, \theta, \mathcal{F}, i)}{1 - E_u(t_{root})} = \sum_{i=1}^{M} q(i) \frac{D_{n,k,i}(t_{root})}{1 - E_{n,k,i}(t_{root})}. \tag{21}$$

Likewise, we can compute the conditional probability density $p(\mathcal{S}_k | \mathcal{T}, \mu, \theta, \mathcal{F})$ of the sequence data at site $k$ given the tree:

$$p(\mathcal{S}_k | \mathcal{T}, \mu, \theta, \mathcal{F}) = \frac{p(\mathcal{T}, \mathcal{S}_k | \mu, \theta, \mathcal{F})}{p(\mathcal{T} | \mu, \theta, \mathcal{F})}. \tag{22}$$

We already know $p(\mathcal{T}, \mathcal{S}_k | \mu, \theta, \mathcal{F})$ from above but now need the tree density $p(\mathcal{T} | \mu, \theta, \mathcal{F})$. This can easily be computed using a birth-death process where the birth rate of each lineage at any time $t$ is always rescaled by its expected fitness $\hat{f}_n(t)$ contained within $\mathcal{F}$.

We can now compute the joint density $p(\mathcal{T}, \mathcal{S}_{1:L} | \mu, \theta)$ for all sites. Because each site is conditionally independent of all other sites given $\mathcal{F}$, we can factor $p(\mathcal{T}, \mathcal{S}_{1:L} | \mu, \theta, \mathcal{F})$ into a product of densities for $\mathcal{S}_k$ at each site and the density of the entire tree $\mathcal{T}$:

$$p(\mathcal{T}, \mathcal{S}_{1:L} | \mu, \theta, \mathcal{F}) = p(\mathcal{T} | \mu, \theta, \mathcal{F}) \prod_{k=1}^{L} p(\mathcal{S}_k | \mathcal{T}, \mu, \theta, \mathcal{F}). \tag{23}$$

We can thus approximate the joint likelihood of the sequence data and the phylogeny $p(\mathcal{T}, \mathcal{S}_{1:L} | \mu, \theta)$ as $p(\mathcal{T}, \mathcal{S}_{1:L} | \mu, \theta, \mathcal{F})$. This allows us to consider how selection shapes sequence evolution at each site while simultaneously considering how the fitness effects of mutations at multiple sites act together to shape the phylogeny. As (*Equation 23*) makes clear though, the goodness of our approximation depends on how well the fitness values in $\mathcal{F}$ are approximated, which in turn depends on how well we can approximate genotypes based on the marginal site probabilities. We explore the goodness of these approximations in the Results section.

## Implementation

We first implemented the marginal fitness birth-death (MFBD) model in Matlab version R2017b. The Matlab implementation was used to test how well the MFBD model can approximate likelihoods and genotype probabilities relative to the exact multi-type birth death model tracking all possible

genotypes for a simple model with only four genotypes. For statistical inference, the MFBD was implemented as an add-on package for BEAST 2 (*Bouckaert et al., 2014*) named Lumière, which extends the existing BDMM package for multi-type birth-death models (*Kühnert et al., 2016*). BEAST two is a general software platform that allows a wide range of evolutionary models including birth-death models to be fit to phylogenetic trees while jointly inferring the phylogeny using Bayesian MCMC sampling. The BEAST 2 implementation of Lumière therefore allows the joint posterior distribution of all parameters in the MFBD model and the phylogeny to be estimated from sequence data. Source code for Lumière and the Matlab implementation are freely available at https://github.com/davidrasm/Lumiere.

## Simulations

To test the statistical performance of our approach, mock phylogenies and sequence data were simulated under a birth-death-mutation-sampling process using a variant of the Gillespie stochastic simulation algorithm (*Gillespie, 2007*) that recorded the ancestry of all individuals in the population. A binary sequence was associated with each lineage and allowed to mutate with a constant per-site mutation rate $\gamma$. Mutations could alter the fitness of a lineage by either increasing or decreasing its birth rate according to site-specific fitness effects. At death events, lineages were sampled with probability $s$, in which case they were included in the mock phylogeny. Code for these simulations is available at https://github.com/davidrasm/Lumiere/tree/master/sim (*Rasmussen, 2019*; copy archived at https://github.com/elifesciences-publications/Lumiere).

## Ebola analysis

We used the Lumière implementation of the MFBD model to estimate the fitness effects of amino acid mutations previously identified to increase the infectivity of Ebola virus in human cell lines (*Diehl et al., 2016*; *Urbanowicz et al., 2016*). We reanalyzed a set of 1610 whole genome EBOV sequences sampled from Guinea, Sierra Leone and Liberia in 2014 to 2016. The sequence alignment along with the time-calibrated molecular phylogeny we used for our analysis were downloaded from https://github.com/ebov/space-time/tree/master/Data.

(*Urbanowicz et al., 2016*) measured the fitness effects of 17 viral genotypes carrying 18 different amino acid mutations in either single, double or triple mutant backgrounds relative to the Makona genotype first sampled at the beginning of the epidemic. Because our methods cannot estimate fitness effects of mutations at very low frequencies, we only analyzed 9 of these mutations that were present in at least 10 of the 1610 viral samples. Preliminary analysis revealed that these mutations fall within eight unique genetic backgrounds because of the way mutations are nested within other single or double mutant lineages in the phylogeny. Because the data of *Urbanowicz et al. (2016)* strongly suggest that epistatic interactions between mutations affect viral fitness, we estimated the genotypic fitness $f_g$ of these eight major genotypes rather than site-specific fitness effects $\sigma$. We therefore used the MFBD to track sequence evolution at each site, but used (*Equation 13*) to marginalize over these genotypes when approximating the fitness of a lineage.

We estimated the fitness of each genotype relative to the Makona genotype, assuming a uniform $[0, 2]$ prior distribution on these fitness values. For the other parameters in the model, we assumed a fixed death or removal rate $d$ of 0.1667 per day based on earlier estimates (*Gire et al., 2014*; *Stadler et al., 2014*). Sampling was modeled as occurring upon removal, with the sampling proportion $s$ set to zero before March 2014, when the first sample was collected. After March 2014, we assumed a fixed sampling proportion of 0.056, reflecting the fact that the dataset included samples from 1610 individuals out of the 28,652 probable cases reported by the WHO (*WHO, 2016*). Lastly, we assumed a constant amino acid mutation rate over all sites with an exponential prior on both the forward and backward mutation rate with a mean rate of $2 \times 10^{-3}$ per site per year. We also ran a second analysis where we included the geographic locations of lineages (Guinea, Sierra Leone and Liberia) as an additional evolving character state in our model. In this analysis, we estimated the effect of geographic location on transmission rates in Sierra Leone and Liberia relative to the base transmission rate in Guinea. Both analyses can be reproduced in Lumière with the XML input files available at https://github.com/davidrasm/Lumiere/tree/master/ebola.

## Influenza H3N2 analysis

We used the Lumière implementation of the MFBD model to estimate the fitness effects of amino acid mutations in the hemagglutinin (HA) protein of human influenza virus subtype H3N2. In order to ensure our fitness estimates were directly comparable to the mutational fitness effects previously estimated by *Lee et al. (2018)*, we focused our analysis on viral samples in the same antigenic cluster as the A/Perth/16/2009 strain studied by Lee et al. for two reasons. First, *Lee et al. (2018)* showed that the fitness effects of amino acid mutations in HA vary depending on the genetic background, with greater fitness differences between more divergent strains. We therefore only considered strains with low genetic divergence from A/Perth/16/2009. Second, the deep mutational scanning experiments were performed in cell culture, and therefore do not reflect the antigenic component of viral fitness in the human population. Only considering a single antigenic cluster therefore minimizes the effect of antigenic mutations.

To further minimize additional background variation in fitness due to geography, we only considered samples collected in the United States from January 2009 to the end of 2012. Overall, we downloaded 2150 sequences from the Influenza Research Database (https://www.fludb.org/) that met these criteria. Nucleotide sequences of the HA segment were aligned in Muscle (*Edgar, 2004*) and a maximum likelihood phylogenetic tree was estimated in RAxML (*Stamatakis, 2014*) using a GTR + Gamma substitution model. To get a time-calibrated phylogeny, branch lengths in the ML tree were converted into units of real calendar time with Least Squares Dating v0.3 (*To et al., 2016*) using a previously estimated molecular clock rate for the HA segment of H3N2 of $5.72 \times 10^{-3}$ substitutions per site per year (*Rambaut et al., 2008*).

In our first analysis, we estimated mutational fitness effects from the H3N2 phylogeny under the MFBD model assuming that fitness effects are multiplicative across sites, as in (*Equation 14*). Because of the large number of naturally occurring mutations in the HA sequences, we limited our analyses to the 17 most abundant amino acid mutations that were present in more than 10% of the sampled sequences. To compare our estimates of population-level fitness effects to fitness effects measured in vitro, we converted the relative amino acid preferences at each site from the deep mutational scanning experiments to mutational fitness effects:

$$\theta_k = log_2 \frac{\pi_{k,i}}{\pi_{k,l}}, \tag{24}$$

where $\pi_{k,i}$ is the relative preference for amino acid $i$ at site $k$. To compute these fitness effects, we used the averaged relative amino acid preferences reported in Dataset S3 of *Lee et al. (2018)*.

In our second analysis, we used the relative preference data from the deep mutational scanning experiments to predict the population-level fitness of viral lineages. For this analysis, we considered all naturally occurring mutations in the HA protein that were present in at least 10 samples. In all, the fitness effects of 67 mutations distributed across 56 sites were included. To map relative amino acid preferences across multiple sites to population-level fitness, we assume that the mutational fitness effects computed from the relative amino acid preferences are additive on a $log_2$ scale, such that the fitness $f_n$ of a lineage is:

$$f_n = (1 + \alpha \sum_{k}^{L} log_2 \frac{\pi_{k,i}}{\pi_{k,l}})^{\kappa}. \tag{25}$$

Here, $\alpha$ is a linear scaling term that allows us to calibrate population-level fitness in terms of the sum of the site-specific fitness effects. We also include the scaling exponent $\kappa$ to account for curvature in the fitness landscape, as might be expected to arise if mutations interact globally through synergistic ($\kappa > 1$) or antagonistic ($\kappa < 1$) epistatic effects across sites (*Elena et al., 2010*). A complete list of the HA mutations considered, their fitness effects predicted by DMS and the XML input file needed to reproduce our analysis are available at https://github.com/davidrasm/Lumiere/tree/master/influenzaH3N2.

## Results

### The four genotype model

We first consider a simple model of molecular evolution in order to compare the marginal fitness birth-death (MFBD) model against the exact multi-type birth-death (MTBD) model tracking all genotypes. Specifically, we consider a binary evolving sequence of length $L = 2$ where all mutations are deleterious and carry a selective fitness cost $\sigma$. Fitness effects of individual mutations act multiplicatively, such that the double mutant has fitness $(1 - \sigma)^2$. With this simple model, it is therefore possible to track the evolutionary dynamics of all four genotypes ($\mathcal{G} = \{00, 01, 10, 11\}$) under both models.

*Figure 2A* shows a phylogeny simulated under the four genotype model, colored according to the genotype of each lineage. We computed the joint likelihood that this tree and observed tip genotypes evolved under a range of different fitness values $\sigma$ for both the exact MTBD and approximate MFBD models (*Figure 2B*). The likelihood profiles under both models peak around the true value of $\sigma$ and closely match at lower values of $\sigma$, but begin to diverge at higher values. The probability of a single hypothetical lineage being in each genotype approximated under the MFBD model is also shown against the exact genotype probabilities computed under the MTBD in (*Figure 2C*).

Because the MFBD approximates the probability of a lineage being in each genotype based on the marginal sites probabilities, we also compared how well the MFBD model approximates the genotype probability densities $D_{n,g}$ relative to the exact multi-type birth-death model. Recall that $D_{n,g}$ provides the probability that the subtree descending from a lineage $n$ has evolved exactly as observed, and therefore forms the foundation of all likelihood calculations under our model. Averaged over all genotypes, the error introduced by approximating $D_{n,g}$ under the MFBD model is greatest at intermediate mutation rates (*Figure 3A*). When there is no selection ($\sigma = 0$), the MFBD introduces no error, but the error increases as the strength of selection increases (*Figure 3B*). We can also consider a variant of the four genotype model where each of the single mutant genotypes is neutral with $\sigma = 0$ but an epistatic interaction between the two sites causes the double mutant to be deleterious with some fitness cost $\epsilon$. Again, the error introduced by the MFBD grows as the strength of the epistatic fitness effect increases (*Figure 3C*).

Taken together, these results suggest that the MFBD model introduces error in $D_{n,g}$ by ignoring correlations among sites due to the fact that selection acts at the level of genotypes, especially when epistasis is strong. The additional correlations between sites induced by selection then causes the genotype probabilities to deviate from those expected based on the marginal site probabilities.

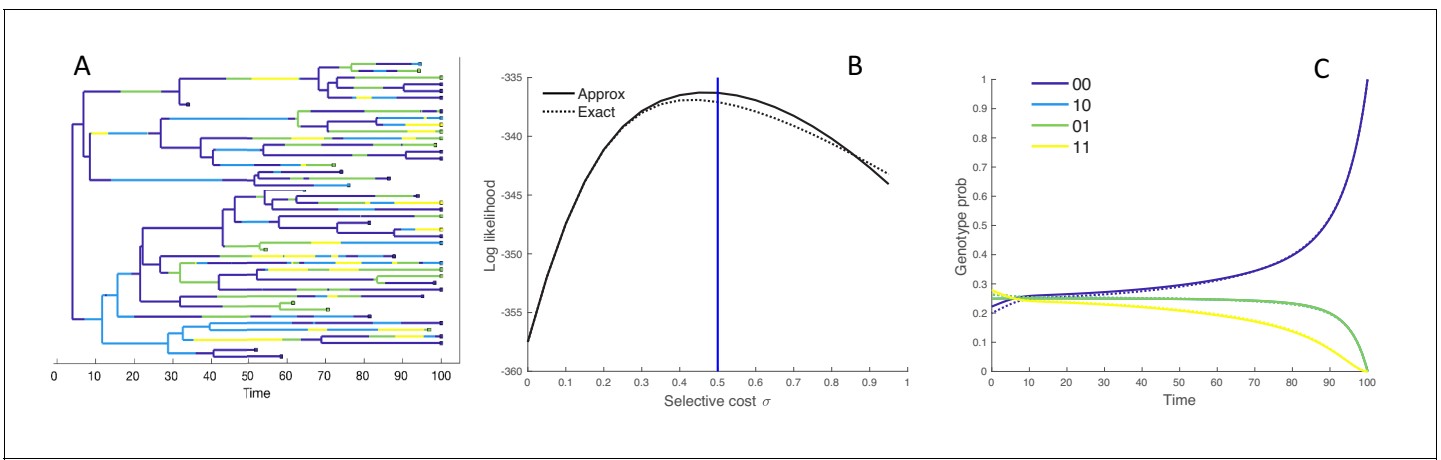

**Figure 2.** Performance of the MFBD approximation under the four genotype model. (**A**) Simulated phylogeny showing the genotype of each lineage through time. (**B**) Joint likelihood of the phylogeny and tip genotypes under different values of $\sigma$ using the the approximate MFBD (solid line) or the exact MTBD model (dotted line). The vertical blue line marks the true parameter value. (**C**) The normalized probability of a single hypothetical lineage being in each genotype back through time based on the MFBD approximation (solid line) versus the exact MTBD model (dotted line) with $\sigma = 0.5$. Note that the probabilities for genotypes 10 and 01 are identical. All parameters besides $\sigma$ were fixed at $\lambda = 0.25$, $d = 0.05$, $s = 0.05$. The mutation rate $\gamma$ was symmetric between forward and backwards mutations and fixed at 0.05.
DOI: https://doi.org/10.7554/eLife.45562.003

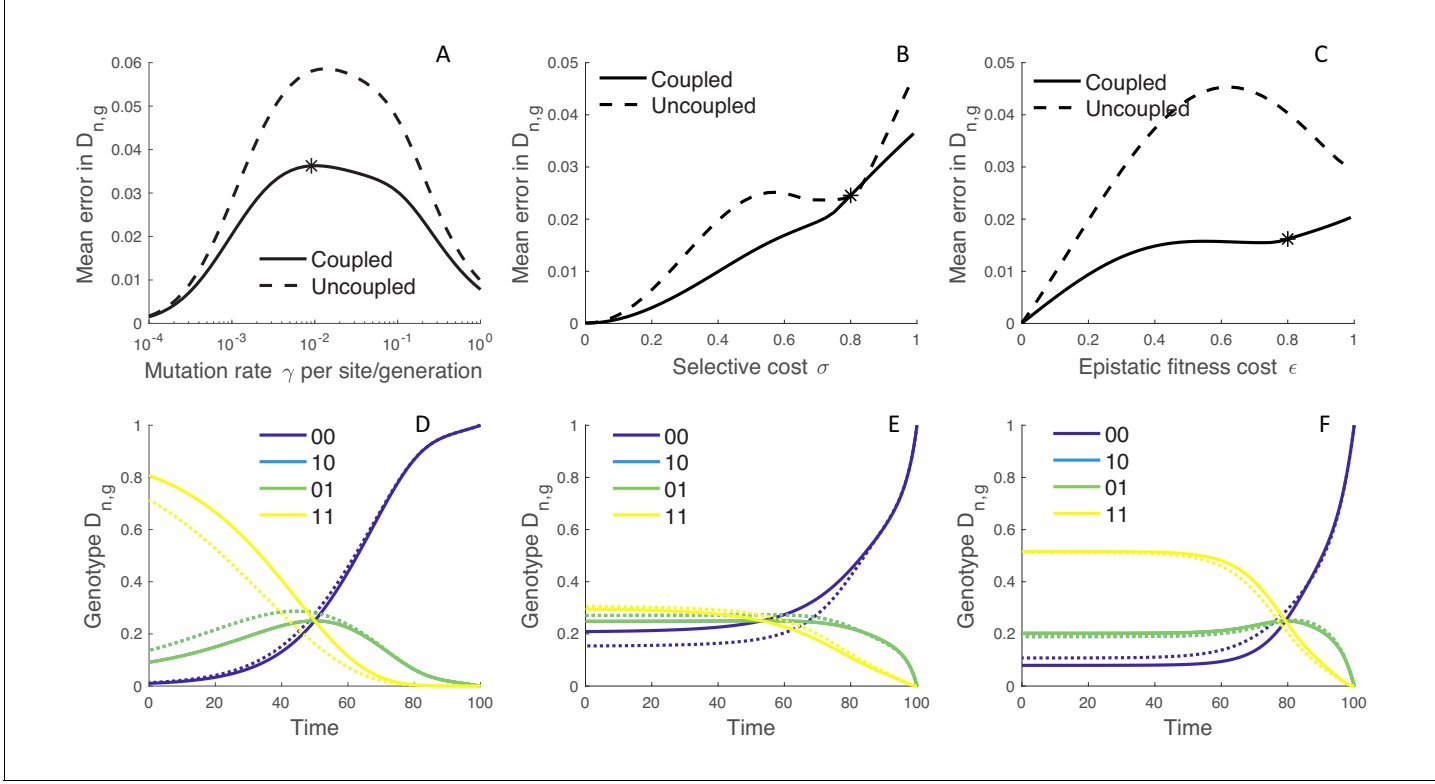

**Figure 3.** The error introduced by approximating genotype probabilities under the MFBD model. (A–C) The error introduced by approximating the genotype probability densities $D_{n,g}$ based on the marginal sites probabilities under the MFBD model for different mutation rates (A), strengths of selection (B), and epistatic fitness effects (C). The solid line represents the MFBD approximation with fitness effects coupled across sites whereas the dashed line represents a more naive approximation that ignores the fitness effects of other sites entirely. The mean error represents the time-integrated average over all genotypes. (D–F) Normalized $D_{n,g}$ probabilities for a single hypothetical lineage being in each genotype back through time based on the MFBD approximation (solid line) versus the exact MTBD model (dotted line). Each plot shows the dynamics of $D_{n,g}$ for the parameter values marked by asterisks in the plots immediately above. Other parameters are fixed at $\lambda = 0.25$, $d = 0.05$, $s = 0.05$.
DOI: https://doi.org/10.7554/eLife.45562.004

Conversely, at very high mutation rates, correlations between sites quickly break down so that sites evolve effectively independently of one another, such that the error introduced by the MFBD also decreases as the mutation rate becomes very high.

Overall, the magnitude of the error introduced by approximating the genotype probabilities is small, especially when we can compare the MFBD model against a more naive approximation that tracks sequence evolution at each site completely independent of all other sites by setting the expected marginal fitness $\hat{f}_{n,k,i} = \sigma$ instead of using (*Equation 14*). This approximation completely ignores how the fitness of a lineage depends on mutations at other sites, and the error in $D_{n,g}$ is generally considerably greater than under the MFBD model (*Figure 3A–C*; dashed lines). Moreover, even when the error introduced by the MFBD model is relatively large, the model still tracks the dynamics of $D_{n,g}$ backwards through time along a lineage well (*Figure 3D–F*; for parameter values marked by the black asterisks in A-C).

The MFBD model also approximates $E_n$, the probability that a lineage has no sampled descendants, using a discretized fitness space and is therefore another source of potential error. Mirroring the results for $D_{n,g}$, the error introduced by this approximation peaks at intermediate mutation rates while it increases monotonically with the strength of selection and epistatic fitness effects (*Figure 4A–C*). Interestingly, tracking how lineages transition between fitness classes in fitness space does not improve the approximation relative to simply ignoring changes in fitness along unobserved lineages (*Figure 4A–C*; dashed lines). The overall magnitude of error introduced by approximating $E_n$ is also small, although using a discretized fitness space does lead to some jaggedness in the

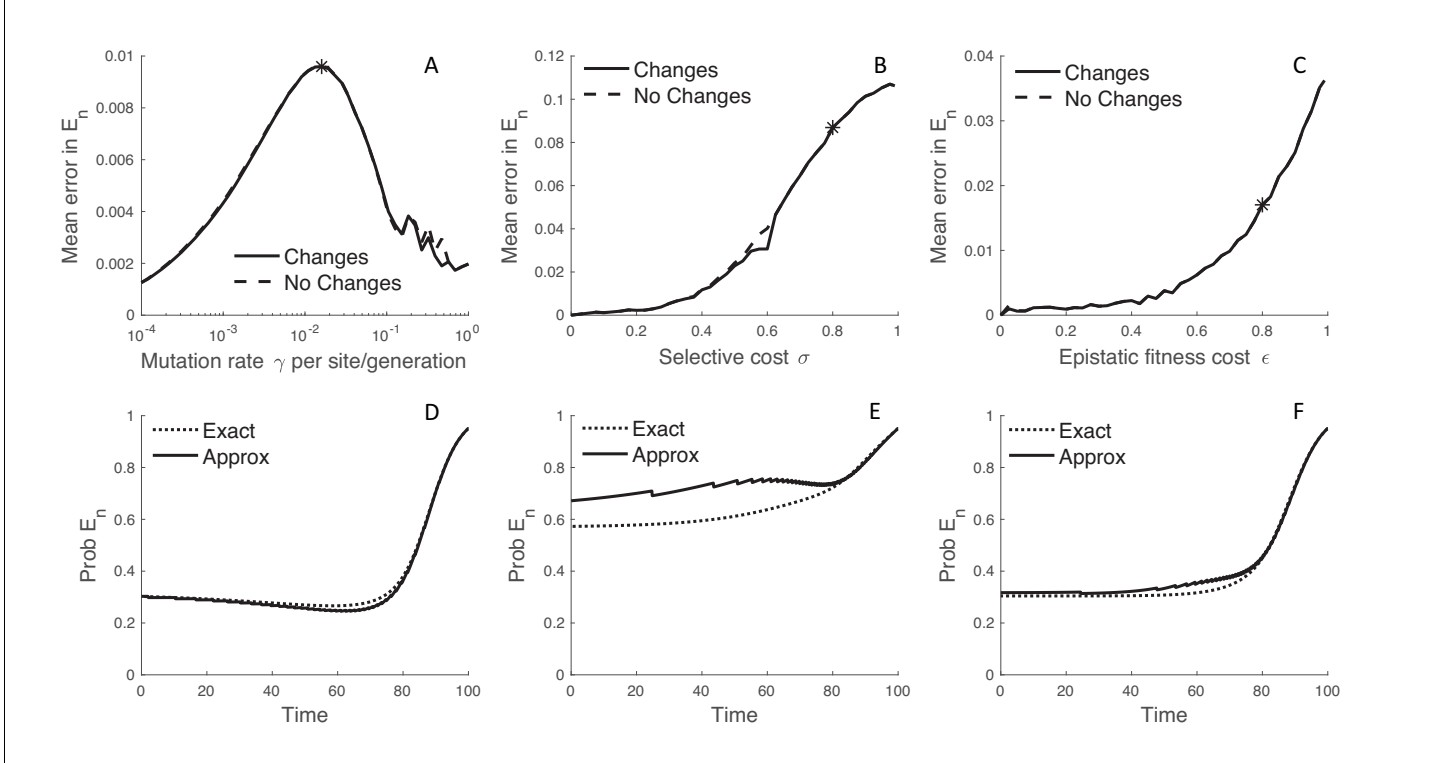

**Figure 4.** The error introduced by approximating the probability of no sampled descendants. (A–C) The error introduced by approximating $E_n$ in a discretized fitness space under the MFBD model for different mutation rates (A), strengths of selection (B), and epistatic fitness effects (C). The solid line represents the approximation where lineages are allowed to transition between fitness classes whereas the dashed line represents the assumption that fitness does not change along unobserved lineages. To obtain a single $E_n$ value comparable across both models, we summed $E_n$ over all genotypes weighted by the exact probability of the lineage being in each genotype and then took the time-integrated average to compute the mean error. (D–F) The dynamics of $E_n$ for a single hypothetical lineage back through time based on the MFBD approximation (solid line) versus the exact MTBD model (dotted line). Each plot shows the dynamics of $E_n$ for the parameter values marked by asterisks in the plots immediately above.

DOI: https://doi.org/10.7554/eLife.45562.005

dynamics of $E_n$ (*Figure 4D–F*). However, only when selection is very strong ($\sigma > 0.8$) does tracking $E_n$ in fitness space result in significant errors, and then only in the more distant past (*Figure 4E*). In this case, a lineage's fitness in the distant past may be a poor predictor of its probability of leaving sampled descendants at a time point in the distant future because the fitness of the lineage and its descendants may greatly change over time in a way that is difficult to predict without considering the exact mutational pathways through which a lineage can move in sequence space.

## Estimating site-specific fitness effects

Next, we simulated phylogenies under a model where the fitness effect of the mutant allele at each site is drawn independently from a distribution of fitness effects (DFE) in order to test how well we can estimate site-specific fitness effects. Because there can be considerable uncertainty surrounding these fitness effects, we now estimate the posterior distribution of fitness effects using Bayesian MCMC. The accuracy and precision of the estimated fitness effects varies considerably across sites, as shown for a representative phylogeny with five evolving sites in *Figure 5*.

In order to better understand this variability, we simulated 100 phylogenies with randomly drawn fitness effects at either 2, 5 or 10 evolving sites. Overall, the estimated posterior median fitness effects are well correlated with their true values, although the strength of this correlation decreases as the number of sites increases (*Figure 6A–C*). Coverage of the 95% credible intervals on the other hand increased from 71.0 to 72.8% to 77.4%.

While there is no systematic directional bias, fitness effects are underestimated for sites at which the mutant allele is at low frequency among sampled individuals and overestimated for sites where

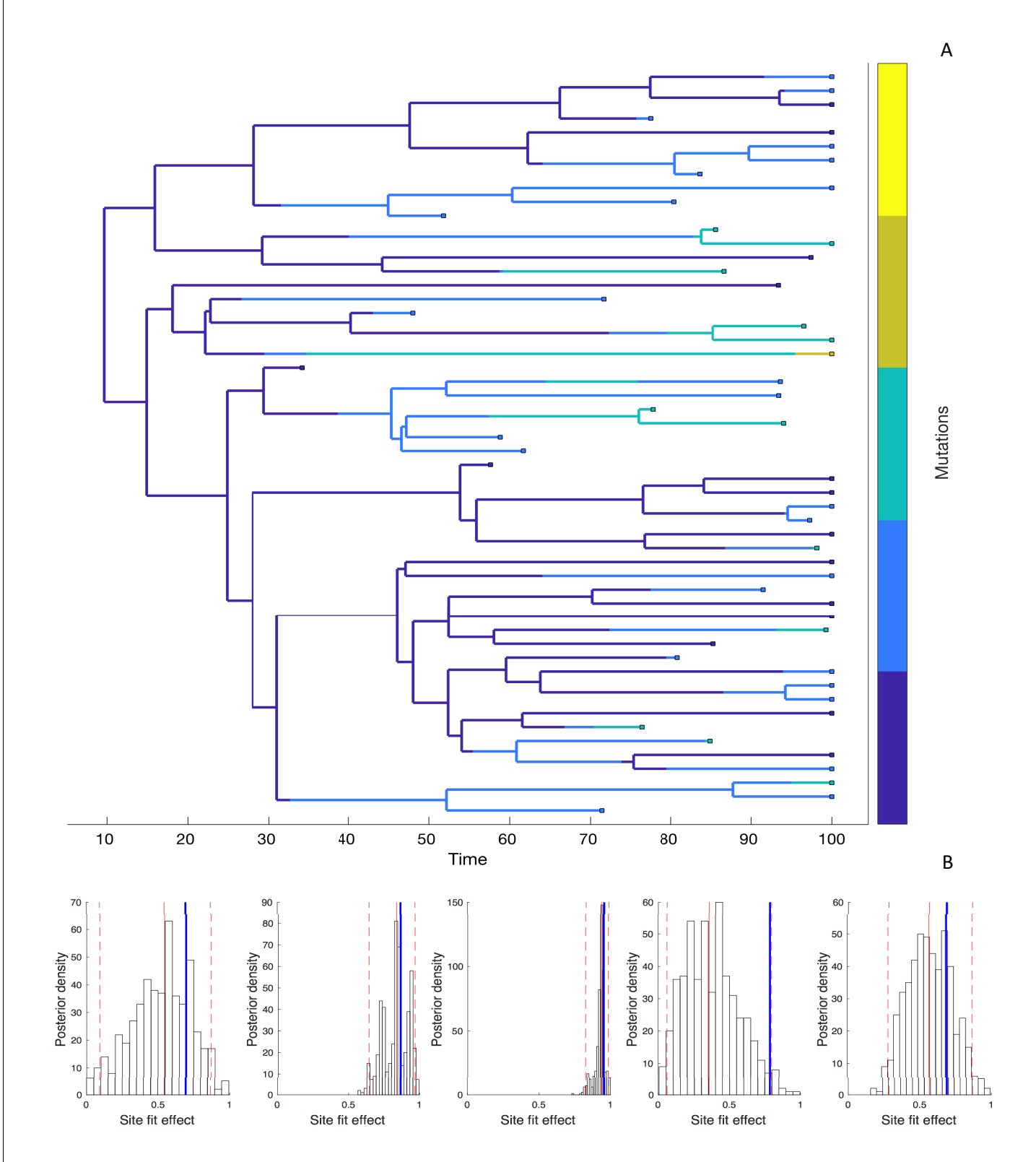

**Figure 5.** Estimating site-specific fitness effects. (**A**) A phylogeny simulated under a model with five evolving sites each with a random fitness effect. The lineages are colored according to the number of mutations they carry (blue = 0; yellow = 5). The distribution of fitness effects was assumed to be LogNormal with a mean of 0.85 and a standard deviation of 0.32. (**B**) Site-specific fitness effects estimated using the marginal fitness BD model. Red lines indicate the posterior median and 95% credible intervals. Blue lines mark the true fitness effect at each site.

DOI: https://doi.org/10.7554/eLife.45562.006

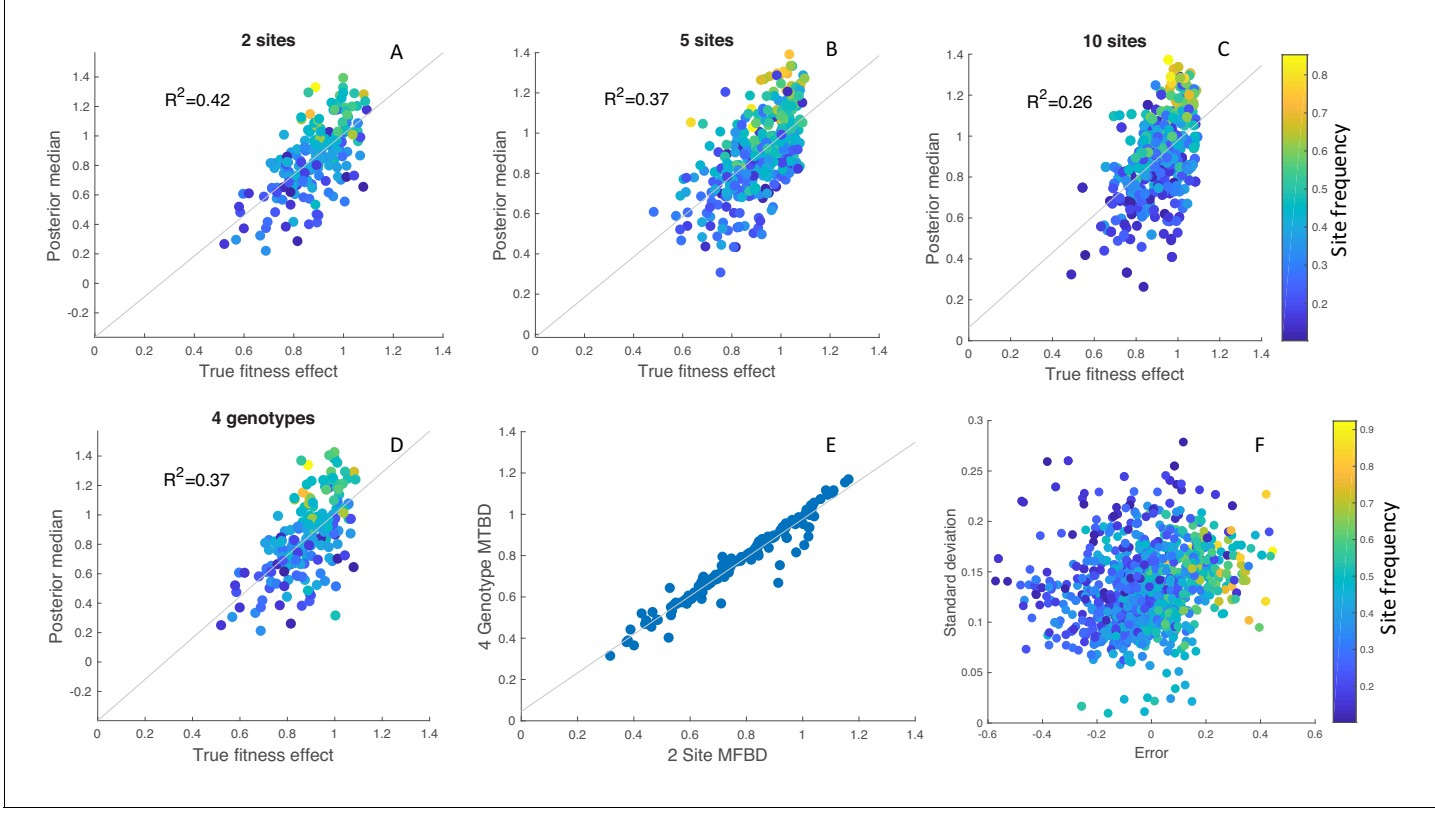

**Figure 6.** Inference of site-specific fitness effects from simulated phylogenies. (**A–C**) Correlation between the true and estimated posterior median fitness effects for phylogenies simulated with 2, 5 or 10 evolving sites. Results are aggregated over 100 simulated phylogenies, with each point representing an estimate for a single site and phylogeny. The points are colored according to the frequency of the mutant allele among sampled individuals in the phylogeny. (**D**) Fitness effects estimated under the exact MTBD model tracking all four possible genotypes for the same two site simulations as in A. (**E**) Correlation between the site-specific fitness effects estimated under the approximate MFBD and exact MTBD for the two site simulations. (**F**) Error and uncertainty in estimated site-specific fitness effects across all 2, 5, and 10 site simulations. Error was calculated as the posterior median estimate minus the true fitness effect. Uncertainty was calculated as the standard deviation of the posterior values sampled via MCMC. In all simulations, sites where the Effective Sample Size of the MCMC samples was below 100 (less than 5% of all sites across simulations) were discarded. The death rate was fixed at $d = 0.05$ but the birth, mutation and sampling rates were randomly drawn for each simulation from a prior distribution: $\lambda$ Uniform(0.1,0.2); $\gamma$ Exponential(0.01); $s$ Uniform(0,1). Only the birth rate was jointly inferred with the site-specific fitness effects.

DOI: https://doi.org/10.7554/eLife.45562.007

the mutant allele is at high frequencies. This however appears to be an intrinsic feature of estimating fitness effects from the branching structure of a phylogeny, as the same phenomena is observed under the exact MTBD model with two sites and four genotypes (*Figure 6D*), and the estimates made under the approximate MFBD model are highly correlated with estimates made under the exact MTBD model (*Figure 6E*).

Across all sites and simulations, accuracy decreased when the mutant allele at a given site was at low or high frequencies, and there was considerably more uncertainty for sites where the mutant allele was at very low frequencies (*Figure 6F*). Thus, while the MFBD model generally performs well at estimating site-specific fitness effects, the accuracy and precision of these estimates varies greatly depending on the frequency of a given mutation in a phylogeny.

## Ebola virus adaptation to humans

The Ebola virus glycoprotein (GP) binds to cells during viral cell entry and is therefore thought to be a key determinant of viral fitness in different hosts. Previously, (*Urbanowicz et al., 2016*) analyzed a large set of naturally occurring amino acid mutations in the GP isolated from patients during the 2013–16 epidemic in Western Africa. The effect of these GP mutations on fitness were then experimentally determined using infectivity assays in cell culture. Several mutant genotypes dramatically

increased viral infectivity relative to the Makona genotype isolated during the earliest stages of the epidemic. However, the effect of these mutations on viral transmission and fitness at the host population level have not yet been determined. We therefore applied the MFBD model to a large dataset of 1610 Ebola virus (EBOV) genomes sampled during the 2013–16 epidemic to infer the population-level fitness effects of these GP mutations.

We analyzed 9 out of the 18 amino acid mutations analyzed by *Urbanowicz et al. (2016)* that were present in at least 10 of the 1610 viral samples. These nine mutations fall in eight different genetic backgrounds or genotypes (*Figure 7*). Because *Urbanowicz et al. (2016)* found evidence for epistatic interactions between several of these mutations, we estimated the fitness of these eight

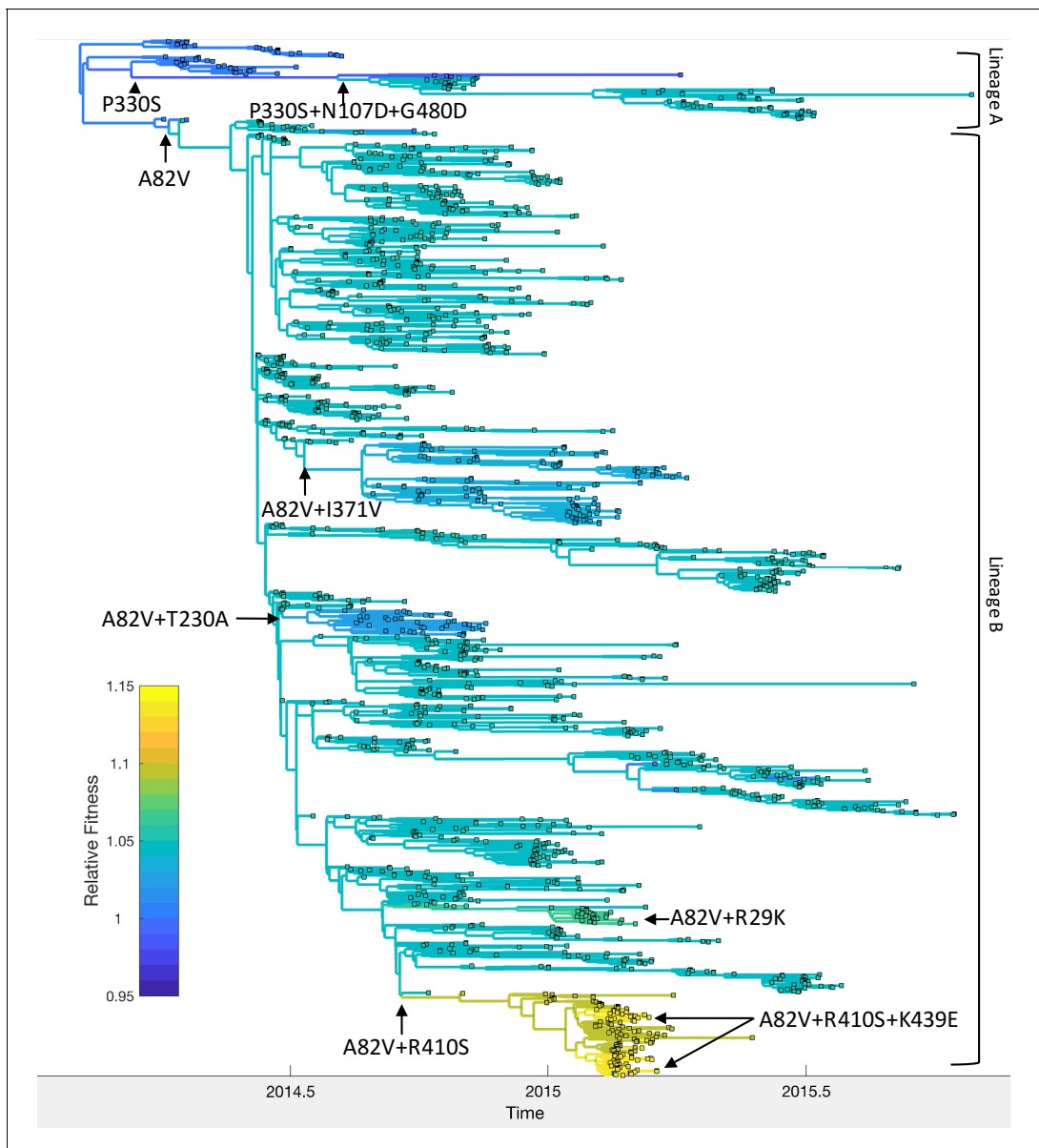

**Figure 7.** Relative fitness of Ebola virus genotypes circulating during the 2013–16 epidemic in Western Africa. Ancestral fitness values were reconstructed by first finding the probability of a lineage being in each possible genotype based on the marginal site probabilities computed using (*Equation 20*). Ancestral fitness values were then computed by averaging the posterior median fitness of each genotype, weighted by the probability that the lineage was in each genotype. Fitness values are given relative to the Makona genotype isolated at the start of the epidemic. Clades are labeled according to their most probable genotype.

DOI: https://doi.org/10.7554/eLife.45562.009

genotypes rather than site-specific mutational fitness effects. *Table 1* shows the relative fitness of these genotypes estimated at the population-level versus their fitness in cell culture.

Mapping the genotypes and fitness of lineages inferred under the MFBD model onto the phylogeny allows us to reconstruct the series of events by which EBOV adapted to humans (*Figure 7*). Shortly after the epidemic started in 2013, the A82V mutation occurred and gave rise to lineage B, which then spread to Sierra Leone, Liberia and Mali. (*Urbanowicz et al., 2016*) found that the A82V mutation increases infectivity by 2–3 fold in cell culture. At the population-level, this mutation appears to have a less dramatic effect, increasing transmissibility by only 5% relative to the Makona genotype. The P330S mutation appears to have temporarily decreased the fitness of the main surviving clade in lineage A, although mutations N107D and G480D later rescue the fitness of this lineage, consistent with the findings of *Urbanowicz et al. (2016)*. Meanwhile, the R410S mutation occurred within lineage B but did not have an immediate effect on fitness. However, R410S appears to epistatically interact with mutation K439E, which occurs twice along the same lineage carrying the R410S mutation and in this genetic background increases infectivity 2–3 fold in cell culture. We estimate that the A82V+R410S+K439E genotype had the highest population-level fitness, but only increased fitness by 14% relative to the Makona genotype. Three other mutations, R29K, T230A and I371V, also occurred in the A82V genetic background, but were not estimated to have further increased the fitness of the A82V genotype.

Because the A82V mutation occurred along a lineage that spread from Guinea to Sierra Leone and several of the genotypes we considered were also geographically restricted, we performed a second analysis to check whether our estimates of genotype fitness were confounded by geographic differences in transmission rates. In this model, we accounted for geographic effects by including location (Guinea, Sierra Leone or Liberia) as an additional evolving character state or 'site' in the model. We found no evidence that transmission rates differed by location; relative transmission rates were 1.01 (95% CI: 0.97–1.05) in Sierra Leone and 0.99 (95% CI: 0.94–1.04) in Liberia compared with Guinea. All mutant genotypes had higher estimated fitness relative to the Makona genotype under the model with geographic effects due to a lower estimated fitness of the Makona genotype. However, the rank order of genotypic fitness values is consistent across models (*Table 1*). Overall, the population level fitness of all eight genotypes agree with their fitness in cell culture in terms of the sign or direction of their effects, but these genotypes had much greater fitness relative to the Makona genotype in cell culture than at the population level.

## Influenza H3N2 fitness variation

We also applied the MFBD model to estimate the fitness effects of mutations in the hemagglutinin (HA) protein of human influenza virus subtype H3N2. *Lee et al. (2018)* recently estimated the relative preference for each amino acid residue at all sites in the HA protein in cell culture using a reverse genetics approach known as deep mutational scanning (DMS). The fitness effect of mutating one amino acid to another is expected to correlate strongly with the relative preference for each amino acid in these experiments. We therefore sought to compare the population-level fitness

**Table 1.** Estimated posterior median fitness and 95% CI for the Ebola GP mutants relative to the Makona genotype

| Genotype | Sample freq | Base model | Model + geo effects | Effect in cell culture |
|---|---|---|---|---|
| Makona | 0.036 | 1.00 | 1.00 | Reference genotype |
| A82V | 0.720 | 1.05 (1.04–1.07) | 1.26 (1.19–1.35) | Increases infectivity 2X |
| P330S | 0.002 | 0.98 (0.82–1.14) | 1.11 (0.96–1.24) | Decreases infectivity |
| P330S+N107D+G480D | 0.037 | 1.04 (0.98–1.12) | 1.27 (1.16–1.39) | Increases infectivity > 2X |
| A82V+R410S | 0.044 | 1.09 (1.00–1.18) | 1.31 (1.17–1.45) | No or small effect |
| A82V+R410S+K439E | 0.035 | 1.14 (1.01–1.26) | 1.36 (1.20–1.54) | Increases infectivity 2-3X |
| A82V+R29K | 0.019 | 1.06 (0.93–1.19) | 1.27 (1.10–1.45) | Increases infectivity 2-3X |
| A82V+T230A | 0.026 | 1.03 (0.93–1.11) | 1.23 (1.10–1.37) | Increases infectivity 2-3X |
| A82V+I371V | 0.067 | 1.03 (0.98–1.09) | 1.24 (1.14–1.35) | Increases infectivity 2-3X |

DOI: https://doi.org/10.7554/eLife.45562.008

effects of naturally occurring mutations estimated under the MFBD model with their fitness measured in vitro by DMS.

To minimize the effect of antigenic mutations, which would not be reflected in the DMS experiments, we limited our analysis to viral lineages in the same antigenic cluster as the A/Perth/16/2009 strain studied by *Lee et al. (2018)*. We first estimated the fitness effects of the 17 most abundant mutations that reached a frequency of 10% or greater among viruses sampled in the United States between 2009 and 2012. We found no apparent relationship between the estimated population-level fitness effects of these mutations and their in vitro effects, although there is agreement that most of these mutations are nearly neutral (*Figure 8A*). Although the 95% credible intervals on our estimates are narrow, these results need to be interpreted with extreme caution because our MCMC algorithm never converged on a stable posterior distribution for several mutations (Effective Sample Size < 10) due to strong correlations between mutations in their estimated fitness effects. This is likely due to the fact that many of these mutations occur only once in the phylogeny and share the same ancestry and therefore genetic background as other mutations in the phylogeny (*Figure 8B*).

While our population-level estimates did not correlate with the in vitro data, the fitness effects predicted by DMS correlate strongly with the maximum frequency that naturally occurring mutations reach in the human population (*Lee et al., 2018*). We therefore sought to test whether using the DMS experimental data to inform the MFBD model about the fitness effects of mutations, rather than estimating them independently from the phylogeny, would result in a better fit of the model to the H3N2 phylogeny and sequence data. Doing so requires a fitness model that aggregates mutational fitness effects across sites and then maps this combined fitness to the population-level fitness of a lineage. In our model, we sum the mutational fitness effects predicted by the relative amino acid preferences across all sites to get a composite predictor of fitness: $\theta_{DMS} = \sum_k^L log_2 \frac{\pi_{k,i}}{\pi_{k,l}}$. We then use (*Equation 25*) to map $\theta_{DMS}$ to overall population-level fitness.

Fitting our model to the H3N2 phylogeny allows us to calibrate how the mutational fitness effects based on relative preferences scale to population-level fitness. Overall, large changes in $\theta_{DMS}$, resulting from mutations to more or less preferred amino acid residues, have a relatively small impact on

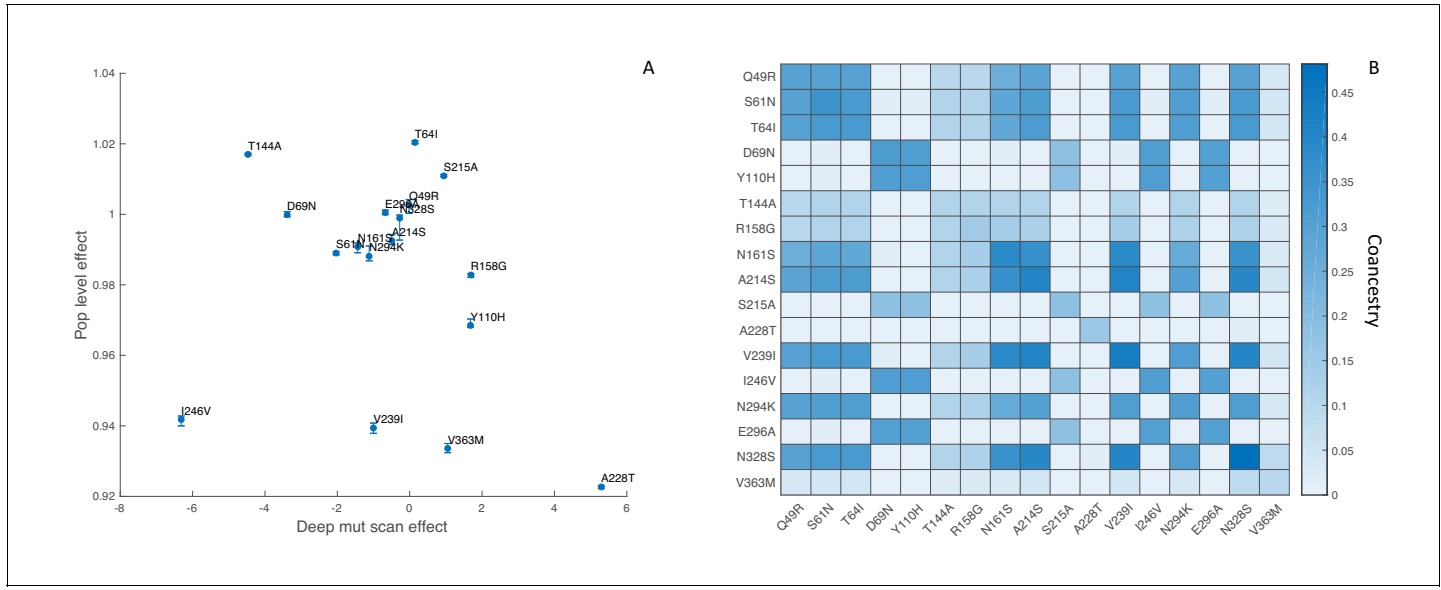

**Figure 8.** Influenza H3N2 mutational fitness effects. (**A**) The fitness effects of mutations estimated in vitro using deep mutational scanning versus their estimated population-level effects. In vitro fitness effects were quantified as the relative preference for the mutant versus the consensus amino acid residue in the deep mutational scanning experiments, given on a log$_2$ scale. Population-level fitness effects were estimated using the MFBD model assuming multiplicative effects across sites. Error bars show the 95% credible intervals on the estimated population-level fitness effects. (**B**) Coancestry matrix showing the fraction of ancestry shared between each pair of mutations in the H3N2 phylogeny. The coancestry value represents the fraction of branches in the phylogeny that share both mutations based on a maximum parsimony reconstruction. The diagonal gives the fraction of all branches in the phylogeny with each individual mutation.
DOI: https://doi.org/10.7554/eLife.45562.010

population-level fitness. Population-level fitness grows slowly and roughly linearly with mutations to more preferred amino acids (*Figure 9*; inset). Nevertheless, when mutational fitness effects are aggregated across all sites, there are substantial fitness differences between lineages (*Figure 9*). Relative to a hypothetical lineage bearing the consensus sequence, fitness ranges from 0.84 to 1.04 across lineages with many lineages having a relative fitness less than one, indicating a slightly deleterious mutation load. Accounting for these fitness differences results in the MFBD model informed by the DMS data fitting the H3N2 phylogeny substantially better (Log likelihood: −4184) than a model assuming all mutations are neutral (Log likelihood: −7510). As would be expected, lineages predicted to be more fit also tend to persist between influenza seasons. Most notably, a lineage with higher than average fitness circulates in 2009 and 2010 during the H1N1 pandemic. This lineages carries the T228A mutation, which is predicted to have a large beneficial effect in the DMS

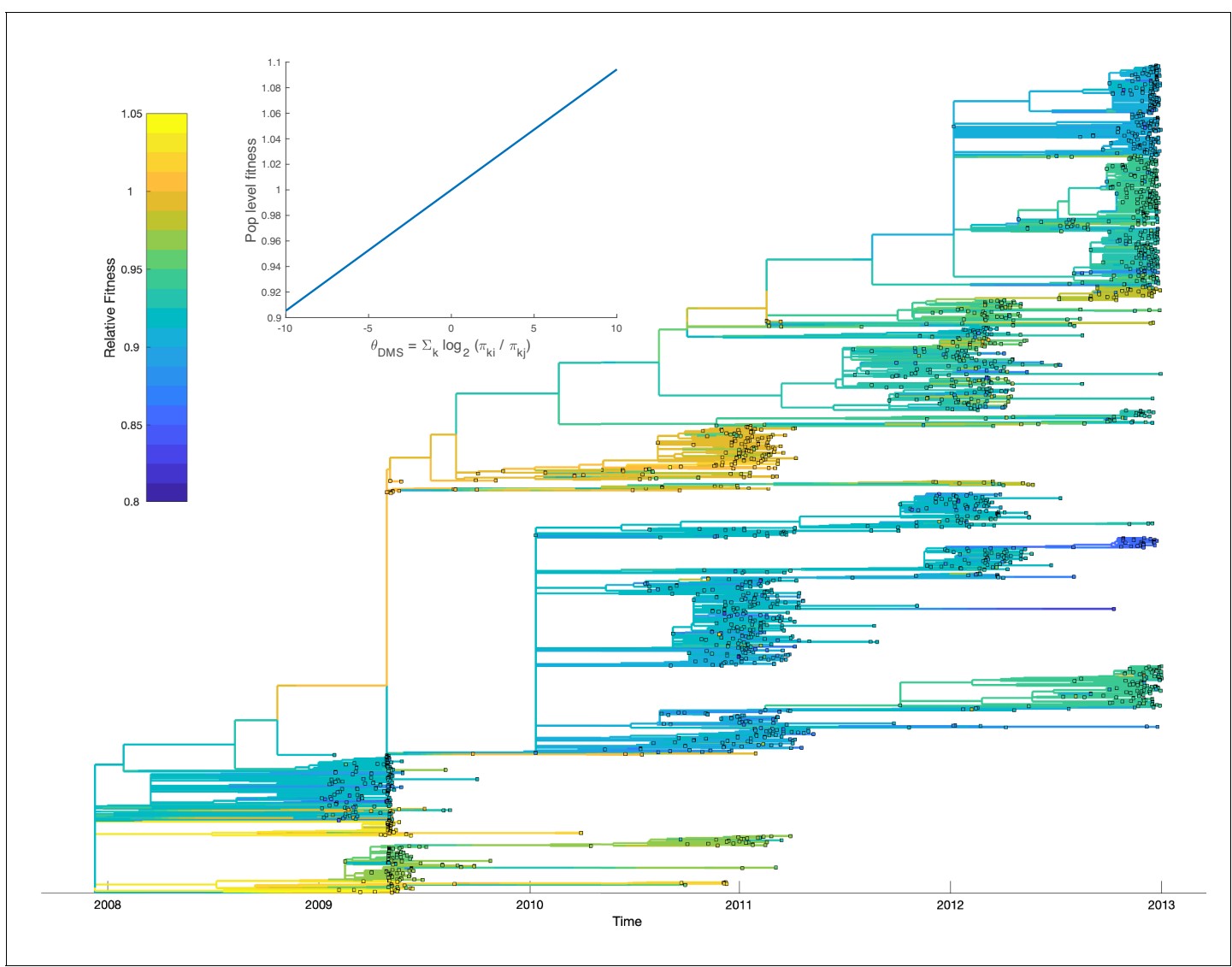

**Figure 9.** Relative fitness of influenza H3N2 lineages circulating in the United States between 2009 and 2012. Fitness values were reconstructed based on a fitness model that maps mutational fitness effects predicted based on deep mutational scanning experiments to population level fitness. The inset shows this fitness mapping for the model parameters with the highest posterior probability: $\alpha = 0.0098$ and $\kappa = 0.964$. Uncertainty in ancestral amino acid sequences was taken into account by first computing the marginal site probability at each site. Ancestral fitness values were then reconstructed by marginalizing over all possible ancestral sequences using the marginal site probabilities.

DOI: https://doi.org/10.7554/eLife.45562.011

experiments. It is therefore tempting to speculate that this mutation may have conferred an advantage that helped seasonal H3N2 compete with the pandemic H1N1 virus.

## Discussion

Many assumptions are made in phylogenetics to model molecular evolution in a statistically tractable way. Historically, one of the most pervasive yet biologically questionable of these assumptions has been that sequences evolve neutrally along lineages, such the mutations do not feedback and alter the branching process shaping the phylogeny. Our marginal fitness birth-death (MFBD) model allows us to relax this core assumption in order to consider how non-neutral evolution at multiple sites affects sequence evolution, the fitness of lineages, and the overall branching structure of a phylogeny. While our approach is not exact in that it approximates genotype probabilities by assuming sites evolve independently when computing the marginal fitness of a lineage, we have shown that this approximation generally works well and only produces significant errors in rather extreme situations, such the four genotype model with very strong selection or epistasis. While an earlier approach based on birth-death models allowed for lineage-specific fitness values to be inferred from the branching pattern of a phylogeny (*Neher et al., 2014*), this approach did not connect fitness back to the mutational process nor allow for the fitness effects of individual mutations or genotypes to be estimated. Using our approach, we demonstrated that the fitness effects of specific mutations can be estimated from simulated phylogenies under the MFBD with accuracy comparable to an exact multi-type birth death model. The MFBD model therefore provides a new, statistically powerful way of incorporating adaptive molecular evolution into phylodynamics.

The MFBD model allows us to exploit phylogenetic information about adaptive evolution that most methods for inferring selection from patterns in sequence data ignore. Currently, codon-substation models (*Goldman and Yang, 1994*; *Muse and Gaut, 1994*) and the related class of mutation-selection models (*Yang and Nielsen, 2008*) are by far the most widely used approach for inferring selection. These approaches rely on comparing sequence substitution patterns such as the dN/dS ratio of non-synonymous to synonymous substitutions across sites. These approaches can be very powerful when sequences from highly divergent taxa are compared, such that enough time has elapsed for multiple substitutions at a single site to have accumulated between lineages. But on the shorter timescales relevant to evolution within a population, substitution patterns like the dN/dS ratio are relatively insensitive to selection pressures and may produce misleading inferences of selection (*Kryazhimskiy and Plotkin, 2008*). For example, a highly beneficial non-synonymous mutation that occurs in a single lineage and then spreads through a population may produce a very low dN/dS ratio, indicative of purifying selection rather than adaptive evolution. In contrast, comparing the evolutionary dynamics of lineages with and without the mutation allows us to infer if that mutation confers a competitive advantage. Thus, considering the branching pattern of phylogenies provides additional information about molecular evolution not visible from substitution patterns in sequence data alone.

While new technologies increasingly allow researchers to quantify mutational fitness effects in vitro or even in vivo (*Zanini and Neher, 2013*; *Thyagarajan and Bloom, 2014*), how fitness measured in the lab translates to fitness in nature is largely unknown. This is especially pertinent for emerging pathogens whose epidemic potential often depends on new adaptive mutations (*Antia et al., 2003*; *Longdon et al., 2014*). Phylodynamic approaches like the MFBD model that can quantify fitness at the host population level are therefore greatly needed, as they offer a means to assess the epidemiological significance of mutant lineages. Extrapolating from our experience with Ebola, where the population-level fitness effects of each mutant genotype we considered matched the sign of their effect in cell culture, we suspect that fitness measured in the lab will generally agree with fitness in nature. This seems reasonable, as mutations that increase replication or cellular infectivity within hosts should generally promote transmissibility between hosts (e.g. *Quinn et al., 2000*; *Fraser et al., 2007*). But at the same time, there is no reason to believe that transmission rates will increase linearly or even monotonically with increasing within-host growth rates. We therefore expect that the magnitude of fitness effects might often greatly differ across scales, as we found for the A82V glycoprotein mutation in Ebola. While A82V doubles infectivity in cell culture, we estimated that it only increases transmissibility at the population level by 5% (95% CI: 4–7%). Interestingly, (*Diehl et al., 2016*) found that A82V only slightly increases viral titers in Ebola patients, which

is likely a much better proxy for transmissibility than cellular infectivity, lending support to our more moderate estimates at the population level.

For influenza, we were unable to reliably estimate the fitness effects of individual mutations from the H3N2 phylogeny. We believe that these inference problems likely stem from the fact that many of these mutations occur only once in the phylogeny and in the same genetic background as other mutations in the HA protein. The shared phylogenetic ancestry of mutations creates an identifiability problem akin to the problem of collinearity in more standard regression-type models. In either case, the individual effects of highly correlated variables are difficult or impossible to infer. Nevertheless, including the mutational fitness effects predicted by deep mutational scanning experiments improved the fit of the MFBD model to the H3N2 phylogeny by thousands of log likelihood units. Accounting for these fitness effects in the MFBD model also revealed substantial variation in population-level fitness among viral lineages within a single antigenic cluster. Most lineages were reconstructed to have a slightly deleterious mutation load, consistent with earlier reports that background variation in fitness arising from deleterious mutations, not just antigenic mutations, plays a large role in determining which H3N2 lineages ultimately persist (*Illingworth and Mustonen, 2012*; *Luksza and Lässig, 2014*; *Koelle and Rasmussen, 2015*). Moreover, the fitness variation uncovered by our analysis likely represents only the 'tip of the iceberg', since there are likely mutations in other genomic segments besides HA with large fitness effects (*Raghwani et al., 2017*), which we did not consider.

The influenza analysis highlights some of the inevitable difficulties encountered when inferring mutational fitness effects from phylogenies. Increasing the number of sites under consideration also increases the complexity of the genetic background in which mutations occur due to the increased probability of mutations being linked to other mutations rather than occurring in isolation. This leads to strong correlations between the fitness effects of different sites in an increasingly high dimensional parameter space, making statistical inference challenging, especially using MCMC methods. Spurious correlations may also arise due to additional, unmodeled sources of fitness variation. For example, if a mutation occurs coincidently with another beneficial mutation or the mutation occurs by chance along a lineage spreading through a higher fitness environment, it will likely be inferred to increase fitness even if it is actually neutral. In the future, the MFBD should therefore be extended to account for unmodeled sources of fitness variation. For example, each lineage could be assigned a random fitness effect representing the unmodeled components of fitness variation. These random effects could then be modeled as a continuous trait evolving along lineages, such that more closely related lineages would be expected to have similar fitness and overly large changes between closely related lineages would be penalized. Such a model would then allow us to say whether a fitness effect attributed to a particular mutation could be equally well explained by random effects arising from unmodeled fitness variation. Until such a principled approach is implemented, the fitness effects of individual mutations need to be interpreted carefully unless they occur in multiple genetic backgrounds and confounding sources of fitness variation can be accounted for, as we tried to do for Ebola by including potentially confounding geographic fitness effects.

In spite of these shortcomings, we believe the MFBD model offers a powerful means to explore many questions not previously possible with strictly neutral phylodynamic models. Even if the fitness effects of individual mutations are not identifiable, it may still be possible to infer the distribution of fitness effects across sites, a key determinant of adaptive evolution that has only been explored in a few systems (*Eyre-Walker and Keightley, 2007*). The MFBD model can also be used to compare the fitness of a mutation or lineage across different environments, such as in different hosts of a pathogen. Finally, the MFBD is not limited to exploring sequence evolution, as the model is generalizable to any discrete character state, including phenotypic, geographic or environmental characters. Thus, more generally, our model can be thought of as a multi-trait, multi-type birth-death model that can be used to explore how different molecular and non-molecular characters interact to shape the overall fitness of lineages in a phylogeny.

## Acknowledgements

We would like to thank Denise Kühnert for helpful advice on how to implement the MFBD model in BEAST 2, Katia Koelle for advice on the influenza H3N2 analysis, and Louis du Plessis for permission to reproduce the model schematic shown in *Figure 1A*.

## Additional information

### Funding

| Funder | Grant reference number | Author |
|---|---|---|
| Seventh Framework Programme | European Research Commission | Tanja Stadler |

The funders had no role in study design, data collection and interpretation, or the decision to submit the work for publication.

### Author contributions

David A Rasmussen, Conceptualization, Software, Formal analysis, Validation, Visualization, Methodology, Writing—original draft, Writing—review and editing; Tanja Stadler, Formal analysis, Supervision, Methodology, Writing—review and editing

### Author ORCIDs

David A Rasmussen https://orcid.org/0000-0001-9457-7561

### Decision letter and Author response

Decision letter https://doi.org/10.7554/eLife.45562.013
Author response https://doi.org/10.7554/eLife.45562.014

## Additional files

### Data availability

All data and code required to reproduce our Ebola analysis in its entirety is available at https://github.com/davidrasm/Lumiere/tree/master/ebola (copy archived at https://github.com/elifesciences-publications/Lumiere). The sequence alignment along with the timecalibrated molecular phylogeny we used for our analysis were downloaded from https://github.com/ ebov/space-time/tree/master/Data. Dataset S3 of Lee et al. 2018 was downloaded from https://www.pnas.org/highwire/filestream/822898/field_highwire_adjunct_files/3/pnas.1806133115.sd03.xlsx.

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
