## [Decision Letter]

Thank you for submitting your article "Coupling adaptive molecular evolution to phylodynamics using fitness-dependent birth-death models" for consideration by *eLife*. Your article has been reviewed by three peer reviewers, and the evaluation has been overseen by a Reviewing Editor and Diethard Tautz as the Senior Editor. The following individuals involved in review of your submission have agreed to reveal their identity: Trevor Bedford (Reviewer #1).

The reviewers have discussed the reviews with one another and the Reviewing Editor has drafted this decision to help you prepare a revised submission.

The reviewers and editor agree that the subject of introducing selection and fitness into phylogenic inference is extremely important, neglected and very hard. For these reasons everyone is very enthusiastic about this effort. Nevertheless, all reviewers raise concerns both about presentation and about validation and applications. The reviewers should be understanding if not all their suggestions can be incorporated but we would urge you to try and satisfy them as much as possible, and very clearly explain instances where you cannot follow their suggestions. In our opinion, given the scope of *eLife* and that many readers will probably be less expert in probability theory than the reviewers, we think it is extremely important to be very clear about the assumptions and where the method works and where it falls short (see comments by reviewer 2). In light of this, reviewer 1 comments that testing different systems will provide such intuition for the less mathematically trained. You do not have to worry about selling the method – I think this is a point where we also learn from negative results.

We are including the reviewers' comments in full, since they are all instructive and each is very different.

*Reviewer #1:*

I've been working in the field of phylodynamics for the last several years, and I can say that it's widely acknowledged how difficult it's been to bridge the more traditional craft of demographic inferences of population size and migration to more fundamental evolutionary insights of strain fitnesses. I applaud the efforts of Rasmussen and Stadler to tackle such an important issue in their manuscript "Coupling adaptive molecular evolution to phylodynamics using fitness-dependent birth-death models". I believe this work represents an innovative modeling avenue to address fitness in phylodynamic inference and will eventually lead to many fruitful biological insights.

I found the model explication clear and well motivated. Although I admit difficulty with the full mathematical details, I could certainly follow the overall approach and was given a good sense of what approximations had been taken and why.

I believe the core modeling advance is completely sound and warrants publication. However, I believe that for *eLife* in particular, a manuscript should point the way towards biological insight. The application to Ebola in West Africa is not as biologically exciting as other systems; the method appears to work, but there's just little adaptive evolution going on. I believe the manuscript would find more traction with readers if it showed a clear example of elucidating impactful fitness differences in an evolving population.

Major suggestions:

1) My central suggestion would be to include a second biological example in which strain variation should be more apparent. Given David Rasmussen's expertise and the known biology of the system, I would suggest H3N2 influenza. Working with influenza phylogenies, there are often clear cut examples of new mutations emerging that rapidly increase in the population (such as HA1 F159Y, https://nextstrain.org/flu/seasonal/h3n2/ha/12y?c=gt-HA1_159). If a few of these genotypes are selected (such as those dictated by the Koel 7, see Koel et al., 2013), are there fitness differences visible in the phylogeny?

I'm not suggesting to do any sort of formal prediction from these fitnesses. Just to see if H3N2 phylogenies give reasonable model outputs.

Alternatively, if flu doesn't work as a motivating example, this should be discussed, as it's so obvious. I don't know if the low sampling fraction would pose an issue to the model.

There were a couple issues I had with Ebola analysis. Nothing fundamental, but issues that should be addressed to get at the biology. These were:

2) I remain concerned about the conflation of epidemiological circumstance and evolutionary fitness in these inferences (Bedford and Malik, 2016, Cell). In particular, the GP A82V mutation corresponds cleanly to a transition from early circulation in Guinea to circulation in Sierra Leone. Could this be addressed technically by adding an additional "site" to the model distinguishing between Guinea, Sierra Leone and Liberia?

3) It seems funny to me to only look the subset of sites investigated by Urbanowicz et al. in the lab. There were other common mutations in GP that weren't looked at by Urbanowicz et al. For example, both R321K and E580G (https://nextstrain.org/ebola?c=gt-GP_321,580) are significantly more common than R29K, I371V and K439E (https://nextstrain.org/ebola?c=gt-GP_29,371,439). Thus, it's weird to me to present a table that lists fitness impacts of several mutations, when there are other mutations that are largely colinear with the presented mutations, that may have actually been the causal variants.

I have this same issue with genes outside of GP. There are highly successful mutations like NP R111C (https://nextstrain.org/ebola?c=gt-NP_111) that are not accounted for. These may be driving the fitness differences observed. For example, L177A visually correlates better with the successful clade than GP E410V, which received a lot of model weight.

Is it possible to put forth a more systematic approach here? I don't think with what's been done that it's possible to conclude much about genetic variants driving transmission differences during the West African epidemic. I worry that handing out an approach as described will lead to people cherry picking what sites they use to define strains in subsequent analyses. There should be some guidance for how to approach this objectively.

*Reviewer #2:*

The manuscript "Coupling adaptive molecular evolution to phylodynamics using fitness-dependent birth-death model" introduces a probabilistic framework to estimate the joint distribution of the sequence data and of the phylogenetic tree, in which the tree is affected by the mutational process. The authors first introduce a general set of coupled different equations for the backwards evolution of the probabilities of observed subtrees and non-observed lineages. As these equations are intractable, a series of approximations to reduce the size of the problem are considered, including projections onto low-dimensional subspaces, decorrelation between sites in the genome… These simplified equations are solved in three cases: two toy models (small number of sites), for which the ground truth is known, and Ebola glycoprotein (GP) data sampled during the 2013-16 epidemic. Based on their computational approach, the authors can quantitatively estimate the fitness effects of 8 GP mutants.

I fully agree with the authors that estimating the likelihood of sequence data and tree altogether, without any unrealistic separation between the tree dynamics and the mutation processes, is an important and very challenging goal. Despite the biological importance of this issue, few works have considered the coupling between the tree dynamics and fitness so far, see for instance Neher, Russel and Shraiman (2014). The present work by Rasmussen and Stadler is an interesting and sophisticated step in this direction. I have however several criticisms against the current version of the manuscript, both on form and content.

My first comment is about the accessibility of the paper, more precisely, whether it is self-contained or not. While some compromise has always to be made between a full, self-contained presentation and the necessity to have a reasonable length, I think the manuscript should be made more accessible. I had to go to Stadler and Bonhoeffer (2013) to understand clearly the meaning of the main quantities (D_n,i_ and E_i_); inserting a figure similar to Figure 1 of that paper would be useful. Equations 2 and 3, which are identical to the ones of Stadler and Bonhoeffer (apart from the presence of mutations – γ_i,j_ – instead of lineage – λ_i,j_ – rates), are also better explained in that paper. Furthermore, the presence of the normalization factor including the probability of being observed in Equation 5 is not really explained here. Again, the 2013 paper was helpful to understand the origin of this denominator.

Secondly, I agree that the set of coupled equations over the D and E variables is intractable, as its solution would require to track an exponentially large (in the genome length) number of quantities. There is no doubt that simplifying approximations are needed, but I am worried that the range of validity of these approximations is not adequately studied here. My concerns are:

1) The dimensionality of the problem is reduced from exponential to linear (in the number of sites) for the D probabilities, while the E probabilities are restricted to a discrete set of points in the fitness space. The corresponding equations, respectively, 19 and 15, are obtained by ad hoc modifications from the original evolution equations, i.e. they are not derived by clear methods that would help the reader understand the magnitude of the terms neglected. It is far from being clear if and when these approximations are acceptable. Again, I understand that approximations are needed, I simply would like to see some justification or some conditions under which the approximations would not affect too much the outcome. The necessity of dimensional reduction is not specific to computational evolutionary biology. Chemical master equations are also virtually of infinite dimensions, and cannot be solved without simplifications, in particular finite-state projections, reminiscent of the procedures followed in the present manuscript. The accuracy of these simplifications is a subject of concern and is studied, see for instance Munsky and Kammash (2006, J. Chem. Phys.).

2) Another drastic hypothesis, besides dimensional reduction, is the decorrelation of sites in the fitness function, see Equation 12. Obviously, epistatic effects are present in reality, and it is therefore important to understand how they would bias the estimates of the fitness effects predicted by the algorithm. This point is studied with the simulated 2-site model (bottom panels in Figure 1), but I was unable to understand what the conclusion (the obvious fact that errors grow with the strength of selection against the double mutant) entails for more complex cases. What can we expect as the genome size grows, and epistatic effects accumulate? Even in the absence of coupling in the fitness, the quality of the prediction of site-fitness effects seems to deteriorate rapidly with the genome size, from 2 to 10 sites, see top panels in Figure 3. The latter size is comparable to the number (9) of mutations studied for GP, but how much could the results of Table 1 affected by the other, not studied mutations in this Ebola virus protein?

3) Across a large phylogenetic tree, it is expected that the relationship between the fitness and the sequences varies across lineages, e.g. owing to changes in the environmental conditions. I wonder how much such effects could influence the predictions for the fitness effects output by the algorithm. To be more specific, suppose that the fitness f changes by some epsilon (depending on the lineage n), how much would the D and E probabilities change? I expect that this could be studied numerically in the small size models, or even analytically with linear-response theory. Assessing the robustness of the predictions against slight variations in the sequence-to-fitness relation is an important issue.

As a conclusion, this work is interesting and can be appreciated as a valuable effort to solve an important problem in phylogeny/sequences estimation. A substantial number of further investigations is, however, required to assess the validity and robustness of the method, before one can fully trust its predictions.

*Reviewer #3:*

This is an interesting paper that addresses the problem of jointly modeling mutation, selection, and sampling from a population, where the individuals are related through a phylogenetic tree. The idea is that selection can alter the expected shape of a tree, and therefore the tree shape is informative about both the genotypic state, and the selection coefficient.

The model builds upon an earlier work (Stadler and Bonhoeffer, 2013) in which a model for the evolution of a single site under selection was introduced. The current paper extends this model to multiple sites under selection. The authors achieve this by making additional assumptions, boiling down to assuming no linkage between these sites.

I am unsure about one aspect of the model – the initialisation of the variable D(t), which is initialized as the product of the probability a lineage is sampled upon death, s, and the rate that the lineage dies, d. I don't see why the death rate comes into this – if one state has a particularly low death rate, it seems that we should be likely to see it, rather than less likely – after all the pathogen does not need to die for the sample to be taken.

The authors show that they are able to infer the selection coefficient / fitness cost in a 4-genotype scenario, where the fitness cost is 0.5. In the application to actual data, the corresponding parameter is inferred to be in the range 5-15%, rather than 50%; and many of the inferences include 0 in their 95% confidence interval. As it stands it is hard to have strong confidence in these results, as there are no simulations to show that the model is sufficiently powered and provides reasonably unbiased estimates in this regime. It would be very helpful if the authors could provide a simulation with parameters in the regime appropriate for the real data.

---

## [Author Response]

Reviewer #1:I've been working in the field of phylodynamics for the last several years, and I can say that it's widely acknowledged how difficult it's been to bridge the more traditional craft of demographic inferences of population size and migration to more fundamental evolutionary insights of strain fitnesses. I applaud the efforts of Rasmussen and Stadler to tackle such an important issue in their manuscript "Coupling adaptive molecular evolution to phylodynamics using fitness-dependent birth-death models". I believe this work represents an innovative modeling avenue to address fitness in phylodynamic inference and will eventually lead to many fruitful biological insights.I found the model explication clear and well motivated. Although I admit difficulty with the full mathematical details, I could certainly follow the overall approach and was given a good sense of what approximations had been taken and why.I believe the core modeling advance is completely sound and warrants publication. However, I believe that for eLife in particular, a manuscript should point the way towards biological insight. The application to Ebola in West Africa is not as biologically exciting as other systems; the method appears to work, but there's just little adaptive evolution going on. I believe the manuscript would find more traction with readers if it showed a clear example of elucidating impactful fitness differences in an evolving population.

Thanks for your comments, Trevor. We acknowledge that Ebola virus has had limited time to adapt to the human population, and other rapidly evolving viruses like influenza exhibit more adaptation.

Major suggestions:1) My central suggestion would be to include a second biological example in which strain variation should be more apparent. Given David Rasmussen's expertise and the known biology of the system, I would suggest H3N2 influenza. Working with influenza phylogenies, there are often clear cut examples of new mutations emerging that rapidly increase in the population (such as HA1 F159Y, https://nextstrain.org/flu/seasonal/h3n2/ha/12y?c=gt-HA1_159). If a few of these genotypes are selected (such as those dictated by the Koel 7, see Koel et al., 2013), are there fitness differences visible in the phylogeny?I'm not suggesting to do any sort of formal prediction from these fitnesses. Just to see if H3N2 phylogenies give reasonable model outputs.Alternatively, if flu doesn't work as a motivating example, this should be discussed, as it's so obvious. I don't know if the low sampling fraction would pose an issue to the model.

We have now also applied our method to influenza H3N2 as a second motivating example. In particular, we applied our method to look at fitness variation among strains in the Perth 2009 antigenic cluster. This allows for direct comparisons of population-level fitness effects estimated using our method with those estimated by Lee et al. (2018) in vitro using deep mutational scanning.

However, we found it very difficult to estimate site-specific mutational fitness effects for H3N2, even when we restricted our analysis to a small subset of naturally occurring mutations. We think that this is likely due to the fact that many of these mutations occur only once in the phylogeny and in the same genetic background as other mutations, and this shared common ancestry between mutations creates a problem with identifiability similar to the problem of co-linearity among highly correlated variables in regression-type models.

While this is disappointing, we also performed a second analysis where we let the deep mutational scanning data from Lee et al. inform our MFBD about the fitness effects of the naturally occurring mutations. This model fit the H3N2 phylogeny dramatically better than a model where mutations were assumed to be neutral. Moreover, reconstructing the ancestral fitness of lineages in the phylogeny under this model revealed substantial fitness variation among lineages within the Perth 2009 cluster, with most lineages carrying a mildly deleterious mutation load.

Given that previous work on H3N2 has argued for the importance of fitness variation within antigenic clusters in determining which lineages survive (Luska and Lässig, 2014; Koelle and Rasmussen, 2015), we believe this application to H3N2 provides another example of how our model can be applied to gain biological insights into the mechanisms of pathogen adaptation.

Looking at the F159Y mutation would have also been be interesting. But we only considered lineages circulating between 2009 and 2012 in our analysis and all lineages circulating before 2014 have the F allele.

There were a couple issues I had with Ebola analysis. Nothing fundamental, but issues that should be addressed to get at the biology. These were:2) I remain concerned about the conflation of epidemiological circumstance and evolutionary fitness in these inferences (Bedford and Malik, 2016, Cell). In particular, the GP A82V mutation corresponds cleanly to a transition from early circulation in Guinea to circulation in Sierra Leone. Could this be addressed technically by adding an additional "site" to the model distinguishing between Guinea, Sierra Leone and Liberia?

This is an excellent point. We have now rerun the Ebola analysis with the geographic location as an additional “site” as suggested. However, we found no differences in transmission rates between locations. The A82V and other mutant genotypes are actually estimated to have slightly larger fitness effects when we account for geographic effects due to a lower estimated fitness for the Makona genotype. The rank order of the genotype fitness values is largely conserved. These results are now included in the main text.

3) It seems funny to me to only look the subset of sites investigated by Urbanowicz et al. in the lab. There were other common mutations in GP that weren't looked at by Urbanowicz et al. For example, both R321K and E580G (https://nextstrain.org/ebola?c=gt-GP_321,580) are significantly more common than R29K, I371V and K439E (https://nextstrain.org/ebola?c=gt-GP_29,371,439). Thus, it's weird to me to present a table that lists fitness impacts of several mutations, when there are other mutations that are largely colinear with the presented mutations, that may have actually been the causal variants.I have this same issue with genes outside of GP. There are highly successful mutations like NP R111C (https://nextstrain.org/ebola?c=gt-NP_111) that are not accounted for. These may be driving the fitness differences observed. For example, L177A visually correlates better with the successful clade than GP E410V, which received a lot of model weight.

While we acknowledge that any of these additional mutations might be driving fitness differences between lineage, we don’t believe including these mutations in our analysis is as interesting as including geographic effects. There will likely always be additional unmodeled sites affecting fitness, but our method is not intended to provide a complete “genome scan” for sites under selection. Rather it is a way to formally test hypotheses about the fitness effects of particular mutations. Thus while our analysis is limited to the mutations studied by Urbanowicz et al., this is the intended use of our method and similar to what we imagine others will use the method for.

Is it possible to put forth a more systematic approach here? I don't think with what's been done that it's possible to conclude much about genetic variants driving transmission differences during the West African epidemic. I worry that handing out an approach as described will lead to people cherry picking what sites they use to define strains in subsequent analyses. There should be some guidance for how to approach this objectively.

We discuss issues with differentiating between the fitness effects of individual mutations versus unmodeled fitness variation in the Discussion. We also warn against over interpreting estimates of mutational fitness effects unless “they occur in multiple genetic backgrounds and confounding sources of fitness variation can be accounted for”. Furthermore, we think that including the additional Ebola analysis with geographic effects provides a good example of how conclusions about the fitness effects of particular mutations can and should be tested by including other potentially confounding sources of fitness variation in the analysis.

However, we agree with the reviewer that there is likely a more general way of accounting for unmodeled fitness effects. We now suggest one potential approach in the Discussion:

“In the future, the MFBD could be extended to account for unmodeled sources of fitness variation. For example, each lineage could be assigned a random fitness effect representing the unmodeled components of fitness variation. These random effects could then be modeled as a continuous trait evolving along lineages, such that more closely related lineages would be expected to have similar fitness due to shared genetic ancestry and overly large changes in fitness between closely related lineages would be penalized. Such a model would then allow us to say whether or not a mutational fitness effect could be equally well explained by random effects arising from unmodeled effects.”

Reviewer #2:[…]My first comment is about the accessibility of the paper, more precisely, whether it is self-contained or not. While some compromise has always to be made between a full, self-contained presentation and the necessity to have a reasonable length, I think the manuscript should be made more accessible. I had to go to Stadler and Bonhoeffer (2013) to understand clearly the meaning of the main quantities (D_n,i_ and E_i_); inserting a figure similar to Figure 1 of that paper would be useful. Equations 2 and 3, which are identical to the ones of Stadler and Bonhoeffer (apart from the presence of mutations – γ_i,j_ – instead of lineage – λ_i,j_ – rates), are also better explained in that paper. Furthermore, the presence of the normalization factor including the probability of being observed in Equation 5 is not really explained here. Again, the 2013 paper was helpful to understand the origin of this denominator.

We have tried our best to make the model description as clear, precise and accessible as possible. But we envision our paper having two types of readers: those more interested in the biological applications and those more interested in the mathematical details. For the first type of reader, we have included a high-level summary of the model in Figure 1. For those interested in the mathematical details, we have chosen to devote more space to describing the new marginal fitness-birth-death model in detail than the original multi-type model. We agree with the reviewer that a reader interested in the full details of the original multi-type model may need to go back and consult Stadler and Bonhoeffer (2013).

We have made adjustments to improve accessibility as the reviewer suggested. Most notably, we have followed the reviewers suggestion and added a panel to Figure 1 illustrating the original multi-type birth-death model (including the D_n,I_ and E_i_ probabilities) and how this differs from the new marginal fitness birth-death model. We also now describe the normalization factor used in Equation 5 to condition the birth-death process on leaving at least on sampled descendent.

Secondly, I agree that the set of coupled equations over the D and E variables is intractable, as its solution would require to track an exponentially large (in the genome length) number of quantities. There is no doubt that simplifying approximations are needed, but I am worried that the range of validity of these approximations is not adequately studied here. My concerns are:1) The dimensionality of the problem is reduced from exponential to linear (in the number of sites) for the D probabilities, while the E probabilities are restricted to a discrete set of points in the fitness space. The corresponding equations, respectively, 19 and 15, are obtained by ad hoc modifications from the original evolution equations, i.e. they are not derived by clear methods that would help the reader understand the magnitude of the terms neglected. It is far from being clear if and when these wild approximations are acceptable. Again, I understand that approximations are needed, I simply would like to see some justification or some conditions under which the approximations would not affect too much the outcome. The necessity of dimensional reduction is not specific to computational evolutionary biology. Chemical master equations are also virtually of infinite dimensions, and cannot be solved without simplifications, in particular finite-state projections, reminiscent of the procedures followed in the present manuscript. The accuracy of these simplifications is a subject of concern and is studied, see for instance Munsky and Kammash (2006, J. Chem. Phys.).

We agree with the reviewer that we gave too little attention to the validity of our approximations under different conditions in the original manuscript. Thus, instead of focusing our analysis of the four-genotype model on individual simulated phylogenies, we now more broadly explore how well the MFBD performs against the exact multi-type birth-death model under different evolutionary dynamics including over a wide range of mutation rates, selection coefficients and epistatic fitness effects. In particular, Figures 3 and 4 now compare the D and E probabilities along a lineage approximated under the MFBD model with the exact model. This new analysis shows that the MFBD approximates the D and E probabilities well under most conditions and that only under rather extreme conditions (very strong selection or epistatic interactions) do approximations introduce appreciable error. But even in these extreme cases the overall magnitude of error introduced is rather small.

In the revision, we also now give more attention to the approximation we used to track the E probabilities in discretized fitness space. While the reviewer sees our approximation as “wild”, we show that our approximation tracks the E probabilities well for a given lineage (see new Figure 4). In fact, using an even more dramatic approximation used in previous multi-type birth-death methods (Rabosky et al., 2014; Barido-Sottani et al., 2018), where transitions between states/fitness classes are ignored altogether along unobserved lineages, still provides accurate results. Thus, while neither approximation tracks the detailed dynamics of how unobserved lineages move through genotype space and thus fitness space, this does not seem necessary in order to capture the overall dynamics of how the E probabilities evolve backwards in time, which is all that is necessary for the MFBD model.

2) Another drastic hypothesis, besides dimensional reduction, is the decorrelation of sites in the fitness function, see Equation 12. Obviously, epistatic effects are present in reality, and it is therefore important to understand how they would bias the estimates of the fitness effects predicted by the algorithm. This point is studied with the simulated 2-site model (bottom panels in Figure 1), but I was unable to understand what the conclusion (the obvious fact that errors grow with the strength of selection against the double mutant) entails for more complex cases. What can we expect as the genome size grows, and epistatic effects accumulate? Even in the absence of coupling in the fitness, the quality of the prediction of site-fitness effects seems to deteriorate rapidly with the genome size, from 2 to 10 sites, see top panels in Figure 3. The latter size is comparable to the number (9) of mutations studied for GP, but how much could the results of Table 1 affected by the other, not studied mutations in this Ebola virus protein?

We expect that the between-site correlations we found for the two-site, four-genotype model would persist between any pair of sites, regardless of the total genome length, because we are assuming complete linkage between sites (no recombination). Thus, we always expect the correlations to become stronger as the strength of selection or epistatic fitness effects grows, regardless of genome length.

However, we do not believe these correlations cause the accuracy of our estimated site-specific fitness effects to decline with more sites. A far more likely reason for this decline is that the complexity of the genetic background in which mutations occur increases with more sites. Mutations therefore start to occur in the same genetic background as other mutations, making it difficult to estimate the effects of individual mutations. This leads to more uncertainty and variability in our estimates of site-specific effects, which explains the loss of accuracy with increasing numbers of sites.

3) Across a large phylogenetic tree, it is expected that the relationship between the fitness and the sequences varies across lineages, e.g. owing to changes in the environmental conditions. I wonder how much such effects could influence the predictions for the fitness effects output by the algorithm. To be more specific, suppose that the fitness f changes by some epsilon (depending on the lineage n), how much would the D and E probabilities change? I expect that this could be studied numerically in the small size models, or even analytically with linear-response theory. Assessing the robustness of the predictions against slight variations in the sequence-to-fitness relation is an important issue.

We also expect that the sequence-to-fitness relationship would vary across the phylogeny due to changes in environmental conditions or other unmodeled fitness effects, such as mutations at other sites. One way of assessing the impact of this background fitness variation on our fitness estimates is to directly incorporate potentially confounding variables into the model. Following the suggestion of reviewer #1, we now take this approach in our Ebola analysis by including the geographic location of lineages in our fitness model.

In the Discussion, we also now suggest a more principled approach to dealing with background variation in fitness due to unmodeled effects: random fitness effects could be assigned to individual lineages, allowing us to say whether a fitness effect ascribed to a particular mutation could equally well be explained by random effects representing unmodeled sources of fitness variation.

We could, as the reviewer suggests, also perform an analysis where we add background “noise” into the mapping between sequences and fitness, but it seems reasonable to expect that the quality of our estimates would depend entirely on the magnitude of this noise.

As a conclusion, this work is interesting and can be appreciated as a valuable effort to solve an important problem in phylogeny/sequences estimation. A substantial number of further investigations is, however, required to assess the validity and robustness of the method, before one can fully trust its predictions.Reviewer #3:This is an interesting paper that addresses the problem of jointly modeling mutation, selection, and sampling from a population, where the individuals are related through a phylogenetic tree. The idea is that selection can alter the expected shape of a tree, and therefore the tree shape is informative about both the genotypic state, and the selection coefficient.The model builds upon an earlier work (Stadler and Bonhoeffer, 2013) in which a model for the evolution of a single site under selection was introduced. The current paper extends this model to multiple sites under selection. The authors achieve this by making additional assumptions, boiling down to assuming no linkage between these sites.

Yes, except to clarify we are actually assuming complete linkage between sites because we do not consider recombination. However, along an individual lineage, we are assuming correlations between the state of different sites can be ignored when calculating genotype probabilities. This is actually quite different from assuming no linkage in a standard population genetics model, where recombination would break up correlations between alleles at different sites at the population level. Our assumption is much more restricted: we are only assuming these correlations can be ignored at the level of an individual lineage.

I am unsure about one aspect of the model – the initialisation of the variable D(t), which is initialized as the product of the probability a lineage is sampled upon death, s, and the rate that the lineage dies, d. I don't see why the death rate comes into this – if one state has a particularly low death rate, it seems that we should be likely to see it, rather than less likely – after all the pathogen does not need to die for the sample to be taken.

In our version of the multi-type birth-death model, sampling is assumed to be coupled to death/removal events. A sampled individual must therefore have died to be sampled, and the D(t) probabilities need to reflect this. However, a lineage can die and not be sampled, and in this case the reviewer’s reasoning is correct: a lineage with a lower death rate will have a higher probability of surviving and therefore of being observed at a future sampling event.

The MTBD can be reformulated such that individuals can be sampled independently of death rates, but we would then need to take into account *sampled ancestors* where a sampled lineage can survive and produce other sampled lineages after the time of sampling. This can be done (see Kühnert et al., 2016), but we feel that this is beyond the scope of an already rather long paper.

The authors show that they are able to infer the selection coefficient / fitness cost in a 4-genotype scenario, where the fitness cost is 0.5. In the application to actual data, the corresponding parameter is inferred to be in the range 5-15%, rather than 50%; and many of the inferences include 0 in their 95% confidence interval. As it stands it is hard to have strong confidence in these results, as there are no simulations to show that the model is sufficiently powered and provides reasonably unbiased estimates in this regime. It would be very helpful if the authors could provide a simulation with parameters in the regime appropriate for the real data.

We do consider estimating site-specific fitness effects in the 5-15% range in Figure 6, where the mutational fitness effects in these simulations were chosen to randomly vary between ~0.4 and ~1.2. While these simulations were not exactly calibrated to the epidemiology of Ebola, they do show that our model can consistently estimate site-specific fitness effects over a wide range of birth rates, mutation rates, sampling fractions and even mutational fitness effects.